# Satellite quantification of enhanced methane oxidation applied to the stratospheric plume following Hunga Tonga-Hunga Ha'apai eruption

Maarten M.J.W. van Herpen [1] ✉, Isabelle De Smedt [2], Daphne Meidan[3], Alfonso Saiz-Lopez [3], Matthew S. Johnson [4], Thomas Röckmann [5] & Jos de Laat[6]

Methane is a powerful greenhouse gas whose atmospheric sink remains uncertain, and emerging strategies to enhance its removal will require quantification and monitoring to verify any hypothetical future methane removal. Here we present satellite quantification of enhanced atmospheric methane oxidation, based on TROPOMI observations of a short-lived intermediate in methane oxidation, HCHO. We find a large HCHO enhancement of up to 12 ppb ±10% at 30 km altitude, in the plume from the Hunga Tonga-Hunga Ha'apai eruption, persisting for ten days or more, and also explaining its low BrO levels. Total methane oxidation is 900 ± 220 Mg/day, suggesting at least 330 Gg of volcanic methane was injected into the stratosphere. The observed methane oxidation requires an estimated ongoing primary production of 2-5 Gg Cl per day that appears unexplained by known mechanisms. We show that chlorine production by iron-chloride photochemistry in sulfate-coated volcanic ash is a plausible mechanism, even outside the marine boundary layer. This method of measuring methane loss using formaldehyde can be sufficiently sensitive to quantify the impact of hypothetical future enhanced atmospheric methane oxidation approaches.

Methane ($CH_4$) is a powerful greenhouse gas, the most important after $CO_2$, and currently responsible for 0.5 °C of warming[1]. The Paris agreement to stay below 2 °C of warming can now only be achieved by rapid and large reductions in methane emissions[2]. The worldwide increase in methane emissions originates from fossil fuel sources, waste treatment, rice production, and especially from agriculture (mainly dairy cows)[3]. At the same time, almost half of methane emissions come from natural sources such as wetlands[3]. The atmospheric sink of methane is also not well-quantified and leads to uncertainty

regarding these methane sources calculated via mass balance arguments[3].

Methane is unique in that it is naturally broken down in the atmosphere within about 10 years, converting $CH_4$ into $CO_2$ and $H_2O$. Because methane is 80 times as potent as $CO_2$ over a 20-year timescale, its relatively fast atmospheric breakdown prevents methane from having an even greater climate impact. This also means that the current warming due to methane is caused only by recent emissions. Addressing methane emissions can therefore result in a reduction of

[1]Acacia Impact Innovation BV, Heesch, The Netherlands. [2]Royal Belgian Institute for Space Aeronomy (BIRA-IASB), Uccle, Belgium. [3]Department of Atmospheric Chemistry and Climate, Institute of Physical Chemistry Blas Cabrera, CSIC, Madrid, Spain. [4]Department of Chemistry, University of Copenhagen, Copenhagen, Denmark. [5]Institute for Marine and Atmospheric Research Utrecht, Utrecht University, Utrecht, The Netherlands. [6]Climate Observations Department, Royal Netherlands Meteorological Institute, De Bilt, The Netherlands. ✉e-mail: maarten@acacia-ii.com

global warming within a decade. Unfortunately, not all methane emissions can be mitigated; it is estimated that the maximum methane emission reduction if all technological options are used is 50%[4], and natural methane emissions are rising due to global warming[5,6]. Despite global efforts to reduce methane emissions, such as through the Global Methane Pledge, atmospheric methane concentrations are rising at their fastest rate in over forty years, with record annual increases observed in 2020 and 2021 of 15.2 ± 0.5 ppb and 17.8 ± 0.5 ppb, respectively[7]. By 2023, the five-year growth rate of atmospheric methane reached its highest level on record[7,8].

In addition to methane emissions reduction, a new field of atmospheric methane removal is emerging that may reduce climate risks by artificially accelerating the natural breakdown of methane in the atmosphere[9]. Climate modeling has shown that large-scale atmospheric methane removal has the potential to reduce future temperatures by 0.5 °C[10,11].

If proven to be climate beneficial and cost-effective, open-air approaches likely have the largest potential scale and fastest time to scale compared to reactor-based approaches[9]. For example, researchers have studied the emission of chlorine to the open air, produced from conventional sea-water electrolysis technology[12]. The most studied approach involves iron-based particles that are lofted into the atmosphere to catalytically generate chlorine radicals that oxidize methane[13]. However, open-air approaches are inherently difficult to verify, and there are concerns about a higher risk of unintended consequences, necessitating strong governance approaches[9]. Such governance requires quantification and observations to verify any hypothetical future methane removal.

Recently, a report by the National Academy of Sciences, Engineering, and Medicine[14] considered the need and viable options for atmospheric methane removal. The report identified monitoring, reporting, and verification as a key challenge, and concluded that we currently lack tools for methane removal quantification. Here, we present a satellite-based quantification methodology that responds to this need.

Satellite observations have proven their value for monitoring emission of air pollution and greenhouse gases[15,16], e.g., supporting policy makers in the international drive to reduce global methane emissions. The capacity of satellites to map global concentrations cannot be matched by other means, e.g., the TROPOspheric Monitoring Instrument (TROPOMI) – currently one of the most advanced satellite instruments for atmospheric composition monitoring – is being used to monitor $CH_4$, $NO_2$, CO, HCHO, and $SO_2$ from continental scales down to the scale of oil and gas infrastructure[15,16].

While it is possible to use local observations to quantify atmospheric methane removal[17], satellite-based observations have been shown to be better suited to determining methane emissions[16]. However, due to the low albedo of water, satellites using reflected shortwave infrared cannot monitor methane over oceans where a substantial portion of natural methane oxidation occurs, and where proposed open-system approaches are envisioned to operate.

Here we demonstrate that locally enhanced atmospheric methane oxidation can be detected using TROPOMI observations of the gasphase species that are involved in methane oxidation, in particular formaldehyde (HCHO). These satellite measurements use UV wavelengths and can also be used over the oceans (unlike methane). Each oxidized methane molecule leads to the production of approximately one HCHO molecule, which is a short-lived intermediate yielding CO within a few hours. In the absence of local HCHO sources such as biomass burning, methane oxidation is the main source of HCHO, and the lifetime is limited to a few hours due to photolysis reactions and the reaction with OH and Cl radicals (see Methods).

As a proof-of-concept demonstration, we will apply this method to the Hunga Tonga-Hunga Ha'apai (HTHH) volcanic eruption, in which chlorine activation was observed.

Hunga Tonga-Hunga Ha'apai (HTHH), a submarine volcano in the South Pacific (20.54° S, 175.38° W) that was situated 150 meters below sea level, erupted violently on 15 January 2022. The area of the HTHH eruption is well-covered by satellites, including TROPOMI, EMI, CALIPSO, MSL, GOES-17, and Himawari-8. As a result, many satellite-based studies of the eruption have been conducted, using observations of species such as $SO_2$, $O_3$, BrO, CO, ClO, HCl, $H_2O$, and aerosol optical depth[18–25].

What made the HTHH eruption exceptional is that it lofted material above 30 km to record-breaking heights of ~55 km[18,19]. The blast released hundreds of Gg $SO_2$[20–22], and injected an exceptionally large mass of $H_2O$ into the stratosphere, estimated to be 146 ± 5 Tg, or ~10% of the total stratospheric burden[21]. The amount of $SO_2$ was modest; for comparison, the Mt. Pinatubo (Philippines) eruption in 1991 injected ~20 Tg of $SO_2$ and reached 40 km at its highest point[23]. Recent work by Wu et al. showed that the low emissions of sulfur by HTHH are due to seawater–magma interactions that removed >93% of a total release of 18.8 Tg $SO_2$[24].

Chlorine activation within the stratospheric plume is evident from in-plume ClO enhancement and ozone depletion observed for about 10 days by the Microwave Limb Sounder (MLS) on board the Earth Observing System (EOS) Aura satellite[25]. Using a coupled chemistry-climate model (WACCM6), Zhu et al. found that a volcanic injection of 1.3 Gg of active Cl explained the observed $O_3$ loss and ClO enhancement[25].

## Results
### Satellite data analysis
Note that we will report column density in units of molec/cm² (with /), concentrations in cm⁻³, and rates in cm⁻³ s⁻¹, to clearly distinguish between them.

The main HTHH eruption occurred on 15 Jan 2022 at 17:00 local time, and is first visible in the Jan 16 TROPOMI overpass at 13:30 (20 hours later). Figure 1 shows the TROPOMI HCHO vertical column density (VCD)[26] for Jan 16 compared with other observations taken around the same time by TROPOMI, VIIRS (on board of SNNP), MLS, and geostationary satellites. The Jan 16 observation shows a very strong HCHO enhancement over New Caledonia that is aligned with enhancements in $SO_2$ (also detected by TROPOMI), aerosol optical depth (detected by VIIRS), as well as sulfate aerosol (SA, see Methods on how this is derived from EUMETSAT Volcanic Ash RGB).

Two clouds, C1 and C2, can be distinguished, with C1 at a slightly higher altitude (30–32 km) than C2 (27–29 km) as observed by CALIOP lidar[27]. The alignment in the location and movement of the HCHO, $SO_2$, and particle enhancements implies that the HCHO enhancement is at the same high altitude, which is unusual for HCHO, which normally is mainly present in the troposphere.

There is a very strong correlation between HCHO and $SO_2$ for both C1 and C2 ($r = 0.8$), with a $\Delta$HCHO/$\Delta SO_2$ enhancement ratio that is 4.5 times higher for C1 (see Table 1, we will refer to observations as C$x$_$y$, with $x$ referring to the cloud 1/2 and $y$ referring to the date in Jan-2022). However, while correlation is equally high with sulfate aerosol ($r > 0.8$), the enhancement ratio $\Delta$HCHO/$\Delta$SA is more similar for C1 and C2. This suggests that the HCHO enhancement is more directly related to aerosols and more indirectly to $SO_2$. We note that previous observations also found that $SO_2$:SA ratios differ in different parts of the HHTH plume[28]. We found that $SO_2$ spectral interference affected HCHO observations in C2 by +40 % on the initial day, but had a low impact of +10 % on observations in C1 (see Methods). By Jan 19, $SO_2$ concentrations have dropped by an order of magnitude and no longer affect the spectral fits.

We quantified the total HCHO enhancement (see Methods) by integrating the VCD over the total surface area, and subtracting background HCHO estimated north and south of the aerosol cloud. We also quantified the total HCHO enhancement by multiplying $\Delta$HCHO/

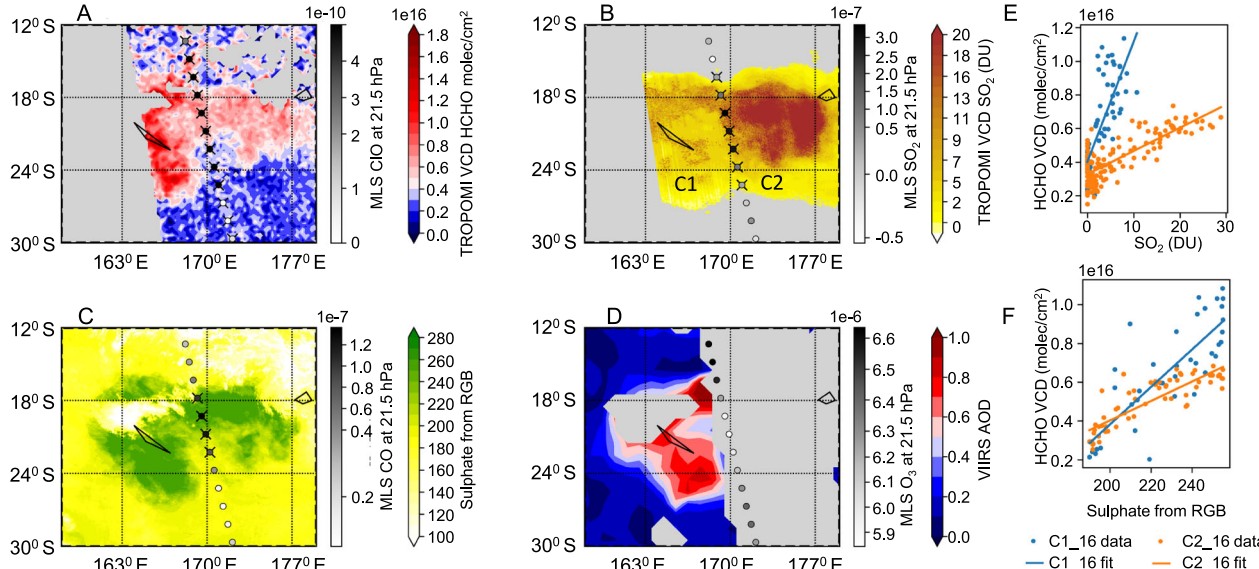

**Fig. 1 | Correlation between HCHO, SO₂, and aerosols within the Hunga Tonga-Hunga Ha'apai plume on 16 Jan 2022. A** HCHO Vertical Column Density (VCD) (without cloud correction) compared with MLS measurements of ClO. MLS data with an X did not pass quality screening. **B** SO₂ column density compared with MLS SO₂ measurements. **C** Modified EUMETSAT Geostationary Ring Volcanic Ash RGB – Multimission (downloaded on 1 May 2025), showing only the green channel to highlight the detection of the sulfate aerosol plume, compared with MLS CO measurements. **D** VIIRS aerosol optical depth (AOD) at 550 nm compared with MLS O₃ measurements. **E** Correlation between SO₂ and HCHO showing different enhancement ratios for cloud C1 and C2. **F** Correlation between sulfate aerosol and HCHO shows a similar enhancement ratio for both clouds.

### Table 1 | Overview of HCHO enhancement ratios that could be confidently assessed

| Observation *** | ΔHCHO/ΔSO₂ (mmol/mol) * | ΔHCHO/ΔSA (molec/cm² per green value) ** | ΔHCHO/ΔAOD (molec/cm² per AOD) * | ΔHCHO/ΔBrO (mol/mol) * |
|---|---|---|---|---|
| C1_16 | 4.5 ± 11% ($r$ = 0.76) | 2.0E + 13 ± 12% ($r$ = 0.81) | 1.3E + 15 ± 08% ($r$ = 0.86) | 41.1 ± 17% ($r$ = 0.62) |
| C1a_19 | 40.0 ± 07% ($r$ = 0.57) | invalid data | invalid data | 12.1 ± 29% ($r$ = 0.17) |
| C1b_20 | SO₂ < DL | 6.8E + 12 ± 22% ($r$ = 0.41) | 4.0E + 14 ± 12% ($r$ = 0.54) | 19.0 ± 22% ($r$ = 0.27) |
| C2_16 | 1.0 ± 06% ($r$ = 0.80) | 1.0E + 13 ± 08% ($r$ = 0.85) | invalid data | 5.5 ± 10% ($r$ = 0.65) |
| C2_17 | 1.1 ± 09% ($r$ = 0.59) | 7.3E + 12 ± 12% ($r$ = 0.61) | 2.2E + 14 ± 16% ($r$ = 0.48) | 6.4 ± 08% ($r$ = 0.62) |
| C2a_20 | 2.2 ± 20% ($r$ = 0.42) | 2.0E + 12 ± 37% ($r$ = 0.32) | 1.7E + 14 ± 27% ($r$ = 0.38) | 6.5 ± 23% ($r$ = 0.37) |
| C2b_21 | SO₂ < DL | 5.3E + 12 ± 14% ($r$ = 0.54) | 3.7E + 14 ± 14% ($r$ = 0.53) | 21.8 ± 11% ($r$ = 0.51) |
| C2b_22 | SO₂ < DL | 3.7E + 12 ± 29% ($r$ = 0.33) | 1.9E + 14 ± 25% ($r$ = 0.35) | 6.4 ± 32% ($r$ = 0.20) |
| C2b_25 | SO₂ < DL | 7.3E + 12 ± 27% ($r$ = 0.31) | 2.6E + 14 ± 15% ($r$ = 0.39) | 10.1 ± 20% ($r$ = 0.24) |
| C3_23 | No correlation ($r$ = -0.04) | invalid data | No correlation ($r$ = -0.07) | No correlation ($r$ = -0.36) |

Showing ± standard error. *SA* Sulfate Aerosol, *AOD* aerosol optical depth, *DL* detection limit.
* after compensating for cloud correction and air mass factor (AMF) error (factor of 4.85)
** Only compensating for AMF error (factor of 4.85).
*** Observations are referred to as Cx_y, with x referring to the observation and y referring to the date in Jan-2022. Coordinates are listed in Table S3.

ΔSO₂ by the total integrated amount of SO₂ and did the same with ΔHCHO/ΔSA. The three quantification methods are in good agreement (see Table 2), showing a total integrated HCHO enhancement of 11.8 ± 2.7 × 10⁶ mol of HCHO for Jan-16 (see Table 2, for clarity we use 1 mol = 6.02 × 10²³ molec).

The highest HCHO VCD enhancement that we observed was 1.6 × 10¹⁵ molec/cm² ± 10% in cloud C1_16 (after sensitivity correction, see Methods). Using a 2 km layer thickness[27] this means the peak HCHO concentration was 8 × 10⁹ molec cm⁻³ ± 10% (12 ppb ± 10% at 20 mbar and 220 K). This stratospheric HCHO concentration is unusually high, with previous observations showing maximum HCHO below 0.1 ppb at 20 km altitude related to biomass burning emissions[29].

Figure 2 shows the HCHO photolysis rates calculated by the Tropospheric Ultraviolet and Visible (TUV) Radiation Model[30], for 25 km altitude at the coordinates of the HCHO enhancement for Jan 15 to Jan 17, scaled to 58% of the TUV rate as modeled by Zhu et al.[25] to represent conditions inside the plume. Based on this HCHO photolysis rate, its

lifetime at around the TROPOMI overpass time (13:30) is ca. 2.5 hours, and any HCHO emitted by the eruption would have been 95% removed by the Jan 16 TROPOMI overpass, and by 99.95% for the Jan 17 overpass. However, the HCHO enhancement remained visible on the following days, indicating continuous HCHO production.

On Jan 17, cloud C2 is located between the Australian coast and New Caledonia (see Figs. 3 and S3), and has a moderately strong correlation with SO₂ and sulfate aerosol ($r$ = 0.6), despite interference from biomass burning HCHO in continental outflow. For C2, the total integrated HCHO enhancement and the ΔHCHO/ΔSO₂ ratio are similar to Jan 16, with a 30 % lower ΔHCHO/ΔSA ratio. We did not quantify the HCHO enhancement for C1 because it is located directly above an Australian biomass burning region and has a high cloud fraction.

On subsequent days, C1 and C2 disperse into elongated shapes that are too large for total integrated HCHO quantification due to interference by clouds and continental HCHO sources. However, we

**Table 2 | Overview of total HCHO enhancements that could be confidently assessed**

| Observation *** | Total HCHO enhancement above baseline (mol)* | Total HCHO via SO₂ correlation (mol) * | Total HCHO via SA correlation (mol) ** | Total HCHO via VIIRS AOD correlation (mol) ** |
|---|---|---|---|---|
| C1_16 | 6.6 E + 06 ± 12% | 6.2E + 06± 11% | 8.4E + 06 ± 12% | 1.1E + 07± 08% |
| C1a_19 | Invalid baseline | 6.6 E + 06± 07% | Invalid data | Invalid data |
| C1b_20 | Invalid baseline | SO₂ < DL | >3.2E + 06 ± 22% | >4.0E + 06± 12% |
| C2_16 | 5.7E + 06 ± 08% | 5.0E + 06± 06% | 3.5E + 06 ± 08% | Invalid data |
| C2_17 | 6.0E + 06 ± 12% | 5.3E + 06± 09% | 5.3E + 06 ± 12% | Invalid data |
| C2a_20 | Invalid baseline | >1.4E + 06± 20% | >6.1E + 05 ± 37% | >1.0E + 06 ± 27% |
| C2b_21 | >3.3E + 06 ± 14% | SO₂ < DL | >3.7E + 06 ± 14% | >3.4E + 06 ± 14% |
| C2b_22 | >2.7E + 06 ± 29% | SO₂ < DL | >1.4E + 06 ± 29% | >1.1E + 06 ± 25% |
| C2b_25 | >5.4E + 06 ± 27% | SO₂ < DL | >1.4E + 06 ± 27% | >4.6E + 06 ± 15% |
| C3_23 | No correlation | No correlation | No correlation | No correlation |

Showing ± standard error. *SA* Sulfate Aerosol, *AOD* aerosol optical depth, *DL* detection limit.
\* After compensating for cloud correction and air mass factor (AMF) error (factor of 4.85).
\*\* only compensating for AMF error (factor of 4.85).
> Means only partial quantification was possible.
\*\*\* Observations are referred to as Cx_y, with x referring to the observation and y referring to the date in Jan-2022. Coordinates are listed in Table S3.

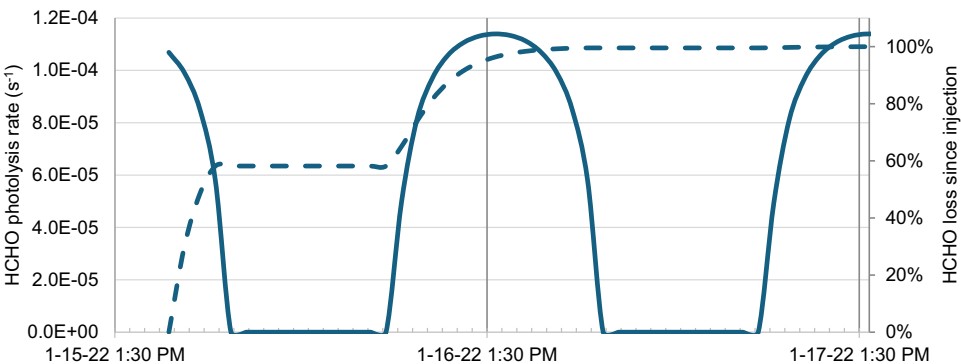

**Fig. 2 | HCHO in-plume photolysis rate as a function of time, starting from the moment of the eruption.** The solid line shows the photolysis rate based on the Tropospheric Ultraviolet and Visible (TUV) Radiation Model[30], scaled to 58% of the TUV rate as modelled by Zhu et al.[25] to represent conditions inside the plume. At the TROPOMI overpass time (13:30, marked with a vertical line), a photolysis rate of 1.14 $\times 10^{-4}\,s^{-1}$ corresponds with a HCHO lifetime of 2.5 hours. The dashed line shows the percentage of HCHO remaining if it were injected by the eruption without in-plume formation, showing 95% HCHO reduction for the Jan 16 TROPOMI overpass, and 99.95% reduction for the second overpass.

still quantified fractions of these clouds and determined their enhancement ratios (see Figs. S2–S19). On Jan 20, cloud C1 was identified over the ocean by the sulfate and AOD correlations, while SO₂ was almost fully removed (Figures S8, S9). Section C1b_20 could be confidently quantified based on its correlation with AOD ($r = 0.54$), and compared to Jan 16, the ΔHCHO/ΔSA and ΔHCHO/ΔAOD enhancement ratios are 70% lower. Nevertheless, at least 36% of the total integrated HCHO enhancement remained compared to Jan 16. For C2 we were able to assess cloud fractions until Jan 25 (Figs. 3 and S18, S19), and found ΔHCHO/ΔSA and ΔHCHO/ΔAOD enhancement ratios that remained stable after Jan 17, while SO₂ became totally removed. For Jan 23, we assessed an SO₂ cloud C3 that was not associated with high AOD or sulfate aerosol, and was found not to contain any HCHO enhancement (Fig. S15).

We analyzed the correlation between TROPOMI BrO and HCHO enhancement (see Table 1), and found a generally linear correlation with ΔHCHO/ΔBrO varying from 6 to 40 mol/mol. The correlation between BrO and HCHO is generally weaker compared to the AOD/HCHO correlation, with a poor BrO/HCHO correlation for cloud C1. For cloud C2, the ΔHCHO/ΔBrO enhancement ratio remained stable between Jan 16 and Jan 25.

We compared the observed HCHO enhancements with coincident simultaneous MLS v5 observations of ClO, SO₂, O₃, CO, H₂O, HO₂ and

HCl (Fig. 3). In the initial days, most of the MLS data did not pass the standard Quality Screening (MLS), which was attributed to extremely enhanced H₂O at very high altitudes[21]. Fortunately, some of the clouds in Table 3 had suitable MLS observations, especially C2b_21 (Figs. 3 and S11).

MLS H₂O is enhanced within the HCHO plume, but the H₂O data generally did not pass quality screening. According to Millán et al., the data can be trusted to show the location of the H₂O enhancement, but the absolute values may have large errors[21]. MLS SO₂ observations were in agreement with TROPOMI observations.

HCHO enhancement was consistently associated with MLS ClO enhancement, but on the first days did not pass quality screening. However, for C2b_21 there was a high-quality MLS measurement for a confident HCHO enhancement (0.7 ppb ClO for C2b_21). We find an O₃ depletion within the ClO/HCHO enhancement, in agreement with previous research[25,31]. For example, O₃ decreased inside the HCHO enhancement from 6.6 to 5.9 ppm in C2_16, from 6.6 to 5.7 ppm in C2_17, and from 6.5 to 6.1 ppm in C2b_21.

CO is elevated within the HCHO plume, with the CO enhancement decreasing over time. The highest enhancement was observed for C2_17, where CO increased from 30 to 130 ppb. CO was elevated by 30 ppb for C1b_20, and by 10 ppb for C2b_21. This was also observed by other researchers and may be due to the combination of an initial

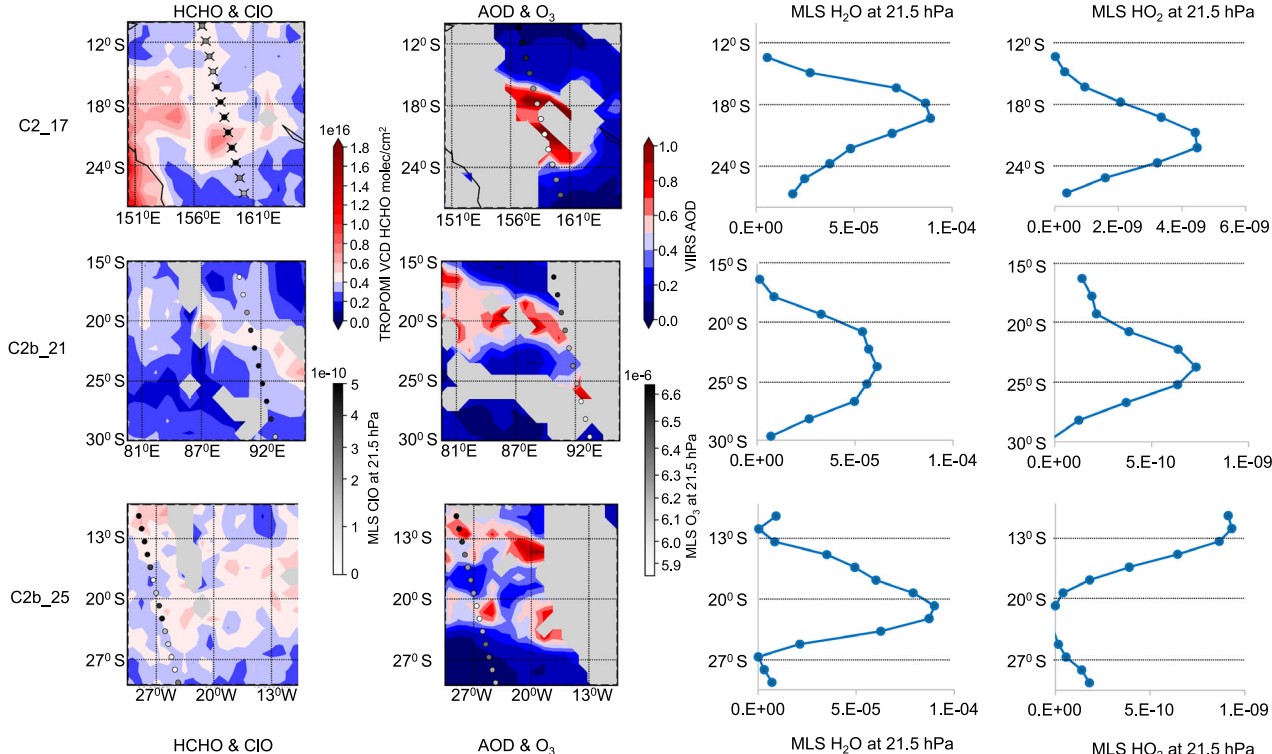

**Fig. 3 | Comparison of several observed HCHO enhancements in vertical column density (VCD) with coincident simultaneous MLS v5 observations.** Showing HCHO (column 1), aerosol optical depth (AOD) (column 2), and MLS observations (columns 3 and 4) for clouds C2_17 (top), C2b_21 (middle), and C2b_25 (bottom). While initially MLS $H_2O$ enhancement is co-located with HCHO/AOD/ $HO_2$ enhancement, by Jan 25, the $H_2O$ enhancement becomes separated from the other enhancements. MLS data marked with an 'X' did not pass quality screening. We did not apply quality screening to MLS $H_2O$, but the data can be trusted to show the location of the $H_2O$ enhancement[21].

updraft of tropospheric air during the eruption, combined with continued CO production in the HTHH plume[25].

For HCl, we find a potential enhancement on Jan 16 and Jan 17, but this data did not pass quality screening. For C2b_21, we found no correlation between HCl and the HCHO enhancement. This is surprising, because other researchers found a slight depletion of HCl compared to the seasonal average[31].

$HO_2$ within the HCHO enhancement is initially strongly elevated (for example, for C2_17 increased to 4.4 ppb from a baseline of 0.5 ppb; on this day, the $HO_2$ data passed quality screening but should still be treated with caution), while later the enhancement is smaller (for example, for C2b_21 we find a 0.5 ppb increase in $HO_2$ within the HCHO enhancement).

## Discussion

The observed HCHO enhancement in the HTHH plume can only be explained by in-plume production. First, the HCHO photolysis lifetime, 2.5 hours at midday, is too short for HCHO emitted by the volcano to remain 20 hours later (see Fig. 2). Second, without in-plume formation, this would reduce HCHO concentrations by 100× between the Jan 16 and Jan 17 overpasses, while on Jan 17, we find a total integrated HCHO enhancement that is approximately the same as for Jan 16. Furthermore, we even detect HCHO enhancements up to Jan 25.

We have observed a linear correlation between HCHO and aerosols (SA and AOD), which suggests that the HCHO lifetime at midday is mainly driven by photolysis (see Methods Eq. 1). If the HCHO lifetime were limited by OH or Cl produced by the aerosols, then the correlation would not be linear (see methods Eq. 2). To be consistent with this, HCHO loss to OH and Cl should be below 50%, resulting in a maximum concentration of $1 \times 10^7 cm^{-3}$ for OH and $2 \times 10^6 cm^{-3}$ for Cl, based on reaction rates (see Table S1 and Fig. S1).

By dividing the HCHO enhancement with a 2.5-hour photolysis lifetime, we derive the total HCHO production rate on Jan 16: $4.7 \pm 1.1 \times 10^6$ mol per hour at midday, with peak values of $9 \times 10^5 cm^{-3} s^{-1} \pm 10\%$ (5 ppb/hour ± 10%). This estimation excludes (minor) HCHO loss to OH and Cl and is therefore a lower estimate. In the absence of other HCHO sources, the $CH_4$ oxidation rate is approximately equal to this amount, amounting to 75±18 Mg $CH_4$/hour at midday. For Jan 20 and Jan 21, we found HCHO enhancements of up to $0.2 \times 10^{15}$ molec/cm² (after sensitivity correction), which corresponds to a concentration of $1 \times 10^9 cm^{-3}$ (1.6 ppb HCHO), and an HCHO production rate of $1.2 \times 10^5 cm^{-3} s^{-1}$ (0.7 ppb/hour).

We calculated a total $CH_4$ oxidation of 900 ± 220 Mg/day (with local peak values of 60 ppb/day on Jan 16 and 8 ppb/day for Jan 20 and 21) in the volcanic plume by assuming the Cl is produced by a photochemical source, using the $NO_2$ photolysis rate to scale the hourly oxidation rate (see Fig. 4). Considering that CO lifetime inside the HTHH plume is reduced to a few days by elevated OH[25], the continued observation of 10-100 ppb CO enhancement in the MLS observations provides further evidence of in-plume production, and fits well with the observed 8 – 60 ppb/day $CH_4$ oxidation. Our observations did not show a decline in the total integrated HCHO enhancement, which suggests this rate of $CH_4$ oxidation may have continued for at least 10 days. This is surprisingly high in view of typical stratospheric $CH_4$ background concentrations of 1 ppm, and suggests that $CH_4$ concentrations in the HTHH plume were elevated.

We calculated the minimum required methane elevation by combining the observed methane oxidation rate with the maximum possible OH and Cl concentrations, and with the known reaction rates for methane oxidation by Cl and OH (see Table S1). On Jan 16, the area of the HCHO enhancement is approximately $7.2 \times 10^6 km^2$ and the thickness is 2 km[27]. This yields an average $CH_4$ oxidation rate of

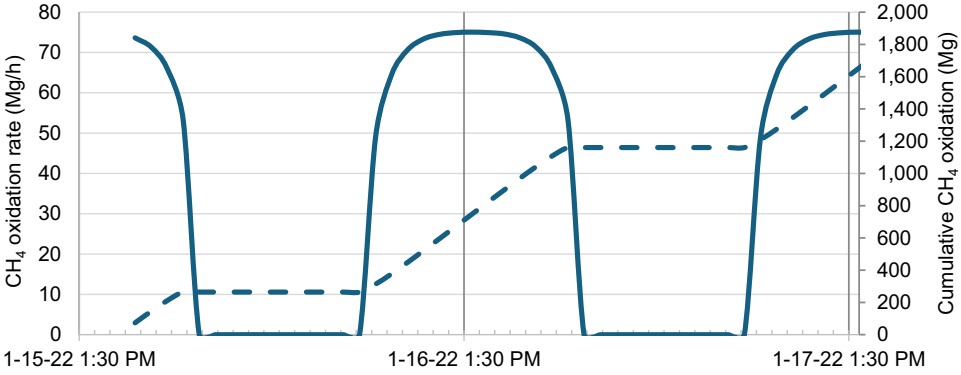

**Fig. 4 | Quantification of total CH₄ oxidation in the Hunga Tonga-Hunga Ha'apai (HHTH) plume.** Showing calculated hourly (solid line) and cumulative (dashed line) CH₄ oxidation in the HHTH plume, based on the observed HCHO production.

$5.5 \times 10^6 \, cm^3 \, s^{-1}$. If 100 % of this is due to a maximum OH enhancement of $1 \times 10^7 \, cm^{-3}$, the $CH_4$ concentration was at least 95 ppm compared to a background value of 1 ppm (an enhancement of at least 2300 Gg $CH_4$). If instead 90 % is due to a maximum Cl enhancement of $2 \times 10^6 \, cm^{-3}$, the $CH_4$ concentration was at least 14 ppm (330 Gg $CH_4$). For explosive eruptions the $CO_2{:}SO_2$ molar ratio can be up to 30[32]. Using an estimated 18.8 Tg of $SO_2$ emission before ocean uptake[24] leads to an estimated $CO_2$ emission of 390 Tg $CO_2$. The range of measured volcanic $CO_2/CH_4$ emission ratios is 10 to $10^5$, with higher values for higher volcanic activity[33], which leads to an estimated $CH_4$ emission of 4 – 40,000 Gg, and fits our estimates based on either OH or Cl.

We examined ACE-FTS data[34] for evidence of $CH_4$ enhancement in the HTHH plume. The earliest ACE-FTS encounter with the HTHH plume occurred 3 weeks after the eruption, on 6 Feb 2022[35]. Unfortunately, due to pointing jumps caused by high aerosol extinction, $CH_4$ observations are invalid for this day. In the monthly average $CH_4$ profiles for 2022, we did not find evidence of a $CH_4$ enhancement that is above the natural swings of 0.05 ppm $CH_4$ with varying altitude (see Fig. S20), while a $H_2O$ enhancement of 7.4 ppm can be seen at 26 km altitude in the Feb 2022 average. The absence of a $CH_4$ enhancement in the ACE-FTS profiles, therefore, implies that the $H_2O{:}CH_4$ ratio in the HTHH emission was less than 150, suggesting the eruption emitted less than 1000 Gg $CH_4$ to the stratosphere. This rules out the possibility that the HCHO enhancement is due to OH (it would mean at least 2300 Gg $CH_4$ emission, which would be clearly visible using ACE-FTS). However, the emission of 330 Gg $CH_4$ due to Cl enhancement is realistic and would indeed not have been detectable with ACE-FTS in Feb 2022.

Thus, the observed HCHO enhancement is due to an increase in Cl, combined with an average methane concentration enhancement of at least 14 ppm in the Jan 16 volcanic cloud. This corresponds to a $CH_4$ vertical column density enhancement of $0.18 \times 10^{19}$ molec/cm², which is around 4% of a typical background measured by TROPOMI, of $4.2 \times 10^{19}$ molec/cm². By Jan 20, the $CH_4$ concentration enhancement is expected to have dispersed by an order of magnitude, in line with the observed lower HCHO concentration enhancement. At this later time, the methane enhancement is therefore also too low to detect with TROPOMI.

The observed HCHO production is sufficiently high that the majority of HCHO could only have been produced by $CH_4$ oxidation (the main precursor for HCHO in the stratosphere), and not from non-methane VOCs (NMVOCs) emitted by the volcano. NMVOCs are known to be emitted by volcanoes, but only at trace concentrations that are at least an order of magnitude less than methane[36]. In addition, the seawater concentrations of DMS (1–10 nM)[37] and dissolved organic carbon (maximum 100 uM)[38] are too low to cause a substantial injection of carbon through the 146 Tg stratospheric $H_2O$ injection.

Zhu et al.[25] used the Whole Atmosphere Community Climate Model version 6 to analyze the chemistry leading to ozone depletion inside the HTHH plume during the first days following the eruption, by constraining the model with MLS observations. For Jan 20 during the daytime, they were able to explain the $O_3$ depletion in the HTHH plume using a mechanism of HOCl uptake resulting in $[Cl] = 6 \times 10^{-14}$ (mixing ratio) $= 4 \times 10^4 \, cm^{-3}$, and a rate of the $Cl + CH_4$ reaction of 900 $cm^{-3} \, s^{-1}$, which is 130× lower than our inferred rate of $1.2 \times 10^5 \, cm^{-3} \, s^{-1}$ (we discuss this mechanism later in the discussion). We also note that the Zhu model found 80 ppt $HO_2$, while MLS observed $HO_2$ concentrations around 500 ppt in the HTHH plume (6× more) on Jan 21. The problem is to explain why the observed HCHO and $HO_2$ concentrations are significantly higher than in the Zhu model output.

We propose that the elevated HCHO and $HO_2$ concentrations arise from the injection of volcanic $CH_4$ into the stratosphere by the HTHH eruption. Increased $CH_4$ causes more Cl to react with $CH_4$ instead of $O_3$, producing HCl and HCHO. This terminates the chain reactions causing $O_3$ depletion in the HTHH plume, such as cycles involving ClO + O, ClO + $NO_2$, and ClO + $HO_2$. Another effect is that enhanced HCHO production leads to enhanced $HO_2$ production, which can produce $O_3$ via reaction with $NO_2$.

A higher primary production of active chlorine is also required to explain the simultaneous loss of $O_3$ and production of HCHO. Based on previous research[39–41] we estimate 0.2 g $CH_4$ is oxidized per g primary Cl produced – accounting for secondary impacts on OH formation and radical chain length. By primary Cl production, we mean the additional Cl that is added to the atmosphere. This leads to an estimated primary Cl production of 375 ± 90 Mg Cl/hour at midday and 4.5 ± 1 Gg Cl/day based on our Jan 16 observations. Peak midday rates are $2.2 \times 10^6 \, cm^{-3} \, s^{-1}$ ± 10 % for Jan 16 and $0.3 \times 10^6 \, cm^{-3} \, s^{-1}$ for Jan 21 (using 0.4 $CH_4$ molecules per Cl atom). This amount may seem similar to the injection of 1.3 Gg ClO used by Zhu[25], but the difference is that to explain both HCHO production and $O_3$ loss, this amount needs to be injected daily.

Bromine chemistry is a key mechanism for chlorine activation in a typical volcanic plume[42,43]. This is a catalytic cycle in which Br activates Cl while depleting ozone. According to Zhu, bromine chemistry cannot explain the observed Cl production in the HHTH plume because it implies a much stronger ozone depletion than was observed[25]. In addition, BrO was observed at a different time during the HTHH eruptions and reached a lower altitude, 8-15 km, where the different wind direction spread the BrO in the opposite southeastward direction compared to the plume that we investigate here[20]. Bromine catalytic cycling is constrained by the Br + HCHO reaction that forms HBr[43]. This shifts bromine speciation towards HBr within our observed strong HCHO enhancements, possibly explaining the relatively low observed BrO compared to $SO_2$ in the high-altitude stratospheric HHTH plume that we investigate, and limiting bromine chemistry as a Cl source. This

also means bromine emissions might have been higher than current BrO-based estimates.

Despite these arguments, we still observe a modest BrO enhancement, and it is correlated with HCHO (see Table 1, and Figs. S2–S19). We calculated the maximum rate of Cl production through bromine chemistry by calculating the rate of formation and reactive uptake of HOBr using observed values for BrO, $HO_2$, and aerosol surface area for cloud C2b_21 (see Supplemental Information Text). We find that the maximum Cl production is $1.5 \times 10^4 \, cm^{-3} \, s^{-1}$, while our observed value is an order of magnitude larger at $3 \times 10^5 \, cm^{-3} \, s^{-1}$. We therefore conclude that Br activation of Cl cannot explain the majority of our observed Cl production.

Previous studies attributed the Cl chemistry to an initial volcanic injection of active Cl, followed by chlorine recycling, especially via ClO + $HO_2$, forming HOCl (see Table S1)[25,31]. Under normal conditions, these recycling mechanisms lead to ozone depletion in which chlorine is catalytic, and it is constrained because there is no production of chlorine to compensate for the loss of active chlorine to $CH_4$ + Cl. However, under the conditions of high aerosol surface area in the HTHH plume, it is possible for chlorine recycling to amplify the total amount of active chlorine (see Fig. S21). The main pathway for this starts with 1 chlorine atom forming HOCl, followed by the reactive uptake reaction of HOCl + HCl that forms $Cl_2$, which photolyzes, yielding two chlorine atoms[31]. We calculated that this mechanism could theoretically reach rates that are high enough to explain some of our observed chlorine production (see Supplemental Information Text). However, the mechanism is driven by the strong dependence of the reactive uptake probability γ on $H_2O$ concentration, which decreases substantially during the days covered by our observations. This mechanism may therefore explain why we observe relatively higher enhancement ratios on Jan 16, but does not explain our observation that the ΔHCHO/ΔAOD enhancement ratios remained stable after Jan 17. We also note that by Jan 25 the $H_2O$ enhancement becomes partly separated from the ClO/$HO_2$ enhancement (see Fig. S19), which does not fit with a Cl source that depends on $H_2O$ concentration, and is a strong argument for why HOCl reactive uptake cannot explain the long-term primary chlorine production implied by our observed HCHO enhancement.

### Iron photochemistry in volcanic ash as a chlorine source

We propose that another possible chlorine source could be iron photochemistry, similar to chlorine production by mineral dust aerosols mixed with sea spray over the North Atlantic[39]. It is estimated that the HTHH eruption released up to 32 kt of iron into the South Pacific Ocean[44]. However, most of the emitted iron is deposited close to the volcano, as is evident from the resulting phytoplankton bloom[45], with only a small fraction reaching the stratosphere. In contrast to initial studies that concluded fine volcanic ash particles were rapidly washed out[27], more recent studies found that fine volcanic ash particles were more likely to remain in suspension, and were difficult to distinguish from more chemically pristine sulfate particles due to a sulfate coating that gave them sulfate-like absorbing properties[46–49].

Romeo et al. used observations and modeling to estimate that between 1.2 and $3.8 \times 10^{11}$ g of fine ash reached the stratospheric cloud[48]. Using an estimated Fe mass fraction of 2–8%[50] leads to an estimated 2.4–30 Gg Fe emission. If we assume 2% of the iron is photoactive (similar to mineral dust)[39], the observed chlorine production of $4.5 \pm 1$ Gg $Cl_2$ per day implies a production rate between 7 and 94 g $Cl_2$ per g photoactive Fe per day. This fits well with the observed value for mineral dust in the marine boundary layer of 70 g $Cl_2$/g Fe per day, especially considering that conditions in the stratosphere are very different, and that sulfate is known to reduce chlorine production in iron photochemistry by up to 40%[51,52].

Using the observed aerosol surface area density of $2.9 \times 10^{-6} \, cm^2 \, cm^{-3}$ for Jan 21[31] combined with an estimated average particle size of 1 μm for coated ash[46] (based on a mix of pristine sulfate aerosols of 0.5 μm and course volcanic ash of 4.6 μm), leads to an estimated aerosol mass of 110 μg/m³, which is reasonable for a translucent plume that is visible on true color satellite images (see Figure S11). Using 50% volcanic ash by weight[46], and again using 2 – 8% Fe mass fraction of which 2% is photoactive, our observed Cl production rate of $0.3 \times 10^6 \, cm^{-3} \, s^{-1}$ for Jan 21 implies a Fe catalytic cycling rate of 1.1–4.5 per hour. Considering the different stratospheric conditions and the presence of sulfate, this corresponds well with the observed value of 11 per hour for mineral dust[39] and values ranging from 6-78 $hr^{-1}$ in laboratory studies[53].

Based on the above, we conclude that iron photochemistry is a plausible source for active chlorine in the HTHH plume. This analysis suggests that iron–chloride photochemistry may be active in the stratosphere, but confirmation will require dedicated modeling and laboratory studies (e.g., a global or plume-resolving model including iron photochemistry and methane injection).

The iron-chloride photochemistry mechanism may not be as significant in other volcanic eruptions, because the HHTH eruption provided unique conditions favorable to iron-chloride photochemistry. This includes the exceptionally large seawater injection that also injected a large amount of sea salt needed for the mechanism. At the same time, the $SO_2$ emission was relatively modest (reducing potential inhibition by sulfate).

### Application of methane removal as a quantification method

We present a methodology for satellite quantification of enhanced atmospheric methane oxidation based on satellite quantification of HCHO, a short-lived intermediate in the $CH_4$ oxidation mechanism. A key advantage is that this approach is especially sensitive to $CH_4$ oxidation, and it works over ocean surfaces where satellite-based $CH_4$ measurements are limited. The use of the methodology is limited by interference from local HCHO sources, but this can be partly overcome through correlations with additional observations such as aerosol optical depth.

When we applied the methodology to the stratospheric plume from the HTHH eruption, we found the highest HCHO enhancement ever recorded in the stratosphere (up to 12 ppb at 30 km altitude), and that the HCHO enhancement persisted for weeks and possibly months. We attributed the HCHO enhancement to a total $CH_4$ oxidation of 900 ± 220 ton/day, with a peak rate of 60 ppb/day on Jan 16. Such a large amount of $CH_4$ oxidation implies that the HTHH eruption must have injected elevated levels of $CH_4$ into the stratosphere.

Meidan et al.[41] modelled local emission of iron for atmospheric methane removal over the ocean and found 25 Gg Cl per hour removed 3.1 Gg $CH_4$ per hour, reducing global radiative forcing by 0.04 W $m^{-2}$ within 10 years. This removal amount is much higher than our observed HHTH removal of 75 ±18 Mg $CH_4$ per hour at midday, which was clearly detectable. Therefore, the sensitivity of our methodology can be sufficient for quantification in hypothetical future enhanced atmospheric methane oxidation approaches to help address future global warming.

## Methods
### Methodology
Our method is to use TROPOMI HCHO observations to quantify enhanced methane oxidation in the HTHH plume, and to use this to derive the required Cl production.

Our hypothesis is that locally enhanced atmospheric methane ($CH_4$) oxidation is revealed using TROPOMI observations of gas-phase species that are involved in $CH_4$ oxidation, in particular formaldehyde (HCHO). $CH_4$ oxidation is mainly initiated by the hydroxyl (OH) radical reaction. In addition, a few percent of $CH_4$ reacts with chlorine atoms (Cl)[54]. Each oxidized $CH_4$ molecule leads to the production of approximately one additional HCHO molecule. HCHO is an

intermediate species that breaks down to form CO within a few hours. CO is a stable intermediate species with a lifetime of about 30 days in the troposphere that eventually oxidizes into $CO_2$.

In the absence of local HCHO sources such as biomass burning, $CH_4$ oxidation is the main source of HCHO. The main sinks for HCHO are two photolysis reactions and its reactions with OH and Cl radicals (see Table S1 and S2).

Using a steady-state approximation, the concentration of HCHO is given by Eq. 1 in which $k_{CH4+OH}$ and $k_{CH4+Cl}$ are the rate coefficients for the reactions of methane with OH and Cl, $k_{HCHO\_OH}$ and $k_{HCHO\_Cl}$ are the rate coefficients for the reaction of HCHO with Cl and OH, and $j_1$ and $j_2$ are the photolysis rate coefficients for HCHO with one channel generating radical (H and CHO) and one molecular ($H_2$ and CO) species.

$$[HCHO] \approx \frac{k_{CH_4+OH}[OH][CH_4] + k_{CH_4+Cl}[Cl][CH_4]}{k_{HCHO+OH}[OH] + k_{HCHO+Cl}[Cl] + j_1 + j_2}$$
$$= \frac{L_{CH_4}}{k_{HCHO+OH}[OH] + k_{HCHO+Cl}[Cl] + j_1 + j_2} \quad (1)$$

In typical marine boundary layer conditions, photolysis is the main sink for HCHO, with $j_1 + j_2$ below $10^{-4} s^{-1}$ [40], depending on season, cloudiness and altitude, while the loss to OH is not insignificant ($5 \times 10^{-5} s^{-1}$ for [OH] of $6 \times 10^6 cm^{-3}$). This implies that for small chlorine enhancements, the concentration of HCHO will increase linearly with the rate of methane oxidation $L_{CH4}$. For higher chlorine concentrations (above $2 \times 10^6 cm^{-3}$), the concentration of HCHO no longer increases linearly, due to increasing loss by reaction with Cl. For extremely high Cl concentrations (above $10^7 cm^{-3}$), both the source and loss of HCHO are dominated by Cl. Here, the concentration of HCHO can be approximated using Eq. 2 with [HCHO] stabilizing at a fixed value that depends on methane concentration instead of oxidation rate. For example, using 1940 ppb $CH_4$, Eq. 2 yields 2.75 ppb HCHO for the marine boundary layer.

$$[HCHO] \approx \frac{k_{CH_4+Cl}[Cl][CH_4]}{k_{HCHO+Cl}[Cl]} = \frac{k_{CH_4+Cl}}{k_{HCHO+Cl}}[CH_4] \quad (2)$$

## Sulfate aerosol (SA) detection

Sulfate aerosol is represented by the green channel of the EUMETSAT Volcanic Ash RGB – that is based on analysis of the infrared channel of several geostationary satellites, and was shown to be correlated with sulfate aerosol in the HTHH plume[27]. In our analysis, we extracted the green channel value, resulting in a number between 0 and 255 that scales with sulfate load.

## HCHO quantification from TROPOMI observation

To quantify HCHO enhancement ($E_{tot}$) in the volcanic plume, we follow this protocol:

1. Quantification by enhancement relative to baseline: in this method, we estimate the enhancement along north-south or east-west transects, by estimating the baseline outside the enhancement region: $E_{tot} = \sum_{i=1}^{N} ([HCHO]_i - [HCHO]_{i,baseline}) A_i$, in which $[HCHO]_i$ is the HCHO VCD for pixel $i$, $[HCHO]_{i,baseline}$ is the baseline value, $A_i$ is the surface area represented by pixel $i$, and $N$ is the number of pixels in the area of interest.

2. Quantification by correlation: in this method, we determine the $\Delta HCHO/\Delta X$ enhancement ratio, in which $X$ may be $SO_2$, SA, or aerosol optical depth. The total enhancement is then the sum of the enhancement in $X$ multiplied by the enhancement ratio: $E_{tot} = \frac{\Delta HCHO}{\Delta X} \sum_{i=1}^{N} (X_i - X_{i,baseline}) A_i$, where $X_{i,baseline}$ is 0 for $SO_2$ and AOD. This methodology makes it easier to quantify the HCHO enhancement in the presence of local HCHO sources (that don't emit species $X$), and provides a more accurate quantification by combining additional data.

## Correction for air mass factor (AMF) in stratospheric HCHO observation

Vertical column density (VCD) represents the amount of a trace gas integrated straight down through the atmosphere, while slant column density (SCD) is the amount measured along the actual, typically angled, light path through the atmosphere. Because the slanted path is longer than the vertical path, the SCD is usually larger than the VCD for the same atmospheric state. The air mass factor (AMF) relates the two by VCD = SCD/AMF. The value of the AMF depends on the length of the light path, the vertical distribution of absorbing trace gases in the atmosphere, the reflectivity (albedo) of the earth's surface, the presence of aerosols and clouds, etc.

The original air mass factor (AMF) that was used to calculate the VCD of HCHO in Figs. 1 and 3 assumed that HCHO is present in the troposphere. However, the enhancement is present in the stratosphere, where the TROPOMI sensor is more sensitive to HCHO, requiring a modified analysis. Instead of an AMF of 1.3 for tropospheric background HCHO, we used the altitude-resolved air mass factors of the HCHO retrievals[55], and took a value for a stratospheric layer around 25 km of 6.3 (correction factor 4.85). In addition, the stratospheric HCHO enhancement is above the clouds, which means it should not be corrected for clouds. To compensate for this, we first determine the HCHO enhancement using cloud-corrected HCHO VCD (as shown in Figs. 1 and 3), and then we apply an AMF correction factor (6.3/1.3) and reverse the cloud correction.

## Role of spectral and aerosol interference for HCHO observations

To check for interference by $SO_2$ in the spectral fits for HCHO we used data from Jan 16 (with the highest $SO_2$ signal), and compared the HCHO slant column density (SCD) fits with and without $SO_2$ (see Fig. S22). Including $SO_2$ in the fits reduces the peak HCHO SCD in C2_16 by about $2 \times 10^{15}$ molec/cm², and about $1 \times 10^{15}$ molec/cm² in C1. Using an AMF of 6.3, this gives a difference in the observed peak VCD of $0.3 \times 10^{15}$ molec/cm² in C2_16, compared to an observed peak enhancement of $0.8 \times 10^{15}$ molec/cm² (40%). There is a difference of $0.15 \times 10^{15}$ molec/cm² in C1_16, compared to an observed enhancement of $1.6 \times 10^{15}$ molec/cm² (10%).

The influence of aerosols on HCHO sensitivity (AMF) is very complicated and depends on aerosol type and altitude relative to the HCHO plume. Both cloud and aerosols can have a shielding effect if the cloud is located above the HCHO plume, but also a reflecting effect if the cloud is located under the plume (in the same way as a bright surface). In addition, sensitivity is generally increased when scattering aerosols (like sulfate aerosols) are vertically collocated with HCHO, while sensitivity is decreased for absorbing aerosols (like volcanic ash) (56). Which effect will dominate depends on the cloud/aerosol properties, and this leads to uncertainties in the stratospheric HCHO observation of the HTHH volcanic plume with an estimated magnitude of $\pm 20\%$[56]. Despite these uncertainties, the detected signal is clear. The observation that $\Delta HCHO/\Delta SA$ and $\Delta HCHO/\Delta AOD$ enhancement ratios remain stable over multiple days, while aerosol scattering/absorbing properties are changing due to $SO_2$ oxidation, provides additional confidence in the detected signal.

In our analysis, we removed the cloud correction for the observed stratospheric HCHO, assuming the stratospheric HCHO is located above clouds and above the majority of the aerosols. This can lead to a slight underestimation of HCHO.

## Data availability

The source data used for Tables 1 & 2 (Supplementary Data 1) and MLS data (Supplementary Data 2) used in the analysis are provided with the paper. The Sentinel 5 P HCHO, $SO_2$, BrO, and AOD data are publicly available through the S5P-PAL Data Portal (https://data-portal.s5p-pal.com/) and the Copernicus Data Space Ecosystem (CDSE) (https://dataspace.copernicus.eu/). MLS v5 data and VIIRS AOD data are

available from the NASA Goddard Space Flight Center Earth Sciences (GES) Data and Information Services Center (DISC) (https://disc.gsfc.nasa.gov/). ACE-FTS data is available through the DatabACE homepage (https://database.scisat.ca/).

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

## Acknowledgements

We thank Chris Boone of the ACE-FTS team for his valuable support in the interpretation of the ACE-FTS data. Part of the study was funded by Spark Climate Solutions (grant numbers AMR-24-007: JdL, AMR-24-009: ID, AMR-24-010: MvH, AMR-23-001: MSJ, AMR-23-002: TR).

## Author contributions

Hypothesis: Mv.H. & Jd.L. Study design: Mv.H., I.D., D.M., A.S.L., M.S.J., T.R., Jd.L. Model simulations: Mv.H. Model analysis: Mv.H. Satellite data analysis: Mv.H., I.D., Jd.L. TROPOMI sensitivity analysis & corrections: ID. Writing – original draft: Mv.H. Writing – review & editing: Mv.H., I.D., D.M., A.S.L., M.S.J., T.R., Jd.L.

## Competing interests

A patent application was filed about the application of satellite-based methane removal quantification to interventions that enhance atmospheric methane removal, which is distinct from the natural atmospheric methane removal enhancement studied here (patent applicant: MvH, inventors: MvH, application number: PCT/NL2026/050017, status: application, receiving office: Netherlands Patent Office / Netherlands). All other authors declare they have no competing interests.
