## [Transparent Peer Review file · Nature Communications]

Satellite quantification of enhanced methane oxidation applied to the stratospheric plume following Hunga Tonga-Hunga Ha'apai eruption

Corresponding Author: Dr Maarten Herpen

Version 0:

Reviewer comments:

Reviewer #1

(Remarks to the Author)

General assessment

The manuscript presents compelling observational evidence of extremely elevated HCHO in the Hunga Tonga eruption plume, and the methodological framework is sound. I agree with the authors' interpretation that heterogeneous activation of chlorine on sulfate aerosols alone cannot account for the inferred chlorine production rates. The proposed contribution from iron photochemistry acting on sulfate-coated ash particles is plausible and consistent with recent advances in our understanding of post-HTHH aerosol composition. The results merit publication after several clarifications and a moderation of some claims.

Major comments

1) Lines 461–462 state that this work provides “the first evidence for iron chloride chemistry occurring outside the marine boundary layer.”

I do not think the current dataset supports this level of certainty. The mechanism is plausible and consistent with the observed HCHO and ClO enhancements, but the attribution remains indirect and the mechanism is not uniquely constrained by the observations presented. A full chemistry–transport evaluation is essential before such a statement can be supported.”

I recommend softening the claim. For example:

“...our analysis suggests that iron–chloride photochemistry may be active in the stratosphere, but confirmation will require dedicated modeling and laboratory studies.”

2) The abstract currently states that “known chlorine production mechanisms cannot explain the high chlorine production rate.”

This should be nuanced. The HTHH eruption is a unique case with exceptional water vapor and aerosol loadings in the stratosphere. Existing parameterizations of heterogeneous chlorine activation were developed for very different regimes and likely require revision before firm conclusions can be drawn. The mechanisms may still be valid, but their parameterizations at extremely high H₂O and aerosol surface area conditions are likely outside their domain of applicability.

I recommend reformulating this to:

“Under the extreme conditions of the HTHH plume, current parameterizations of chlorine activation appear insufficient to account for the inferred production rate.”

3) In the abstract (line 37), the authors state that iron–chloride photochemistry is a plausible mechanism.

Based on the presented correlations and scaling arguments, I agree that the mechanism is plausible. I would, however, avoid calling this “evidence” in the absence of a full chemical box or 3-D model evaluation. The authors may instead say:

“Our analysis supports iron–chloride photochemistry as a strong candidate mechanism.”

4) To substantiate the proposed mechanism, a targeted chemical modeling experiment would be required (e.g., WACCM, CESM, or a plume-resolving model including iron photochemistry). The current observational correlations cannot exclude competing or complementary pathways.

I suggest adding a statement acknowledging this limitation and indicating that model evaluation is needed to test the

hypothesis.

Minor comments

1. the authors might want to add a short paragraph in the method section that explains the difference between VCD, SCD and the assumptions to estimate a concentration. That will help the reader unfamiliar with satellite retrievals
2. The authors should also explain a bit further how the HTHH eruption has affected the MLS and other satellites retrievals. It will help to better understand the discussions behind “quality screening”.
3. It would also be interesting to briefly comment on whether similar iron–chloride photochemical pathways might have been relevant for past eruptions such as Pinatubo, and whether the much lower stratospheric water vapor during that event would have limited or enhanced such chemistry.

Reviewer #2

(Remarks to the Author)

Reviewer report on: Satellite quantification of enhanced methane oxidation applied to the stratospheric plume following Hunga Tonga-Hunga Ha’apai eruption

Key results

In this manuscript, the authors report an analysis of chemistry of the plume of the Hunga Tonga-Hunga Ha’apai (HTHH) eruption of January 2016, which was emitted into the stratosphere. Using data from satellite observations, principally TROPOMI, high amounts of HCHO are shown to be present in two clouds this stratospheric plume. The peak HCHO concentration is determined to be the highest ever observed within the stratosphere.

This HCHO is found to be persistent within the plume – despite HCHO’s expected short lifetime to photolytic destruction. The authors argue is evidence of continual HCHO formation from volcanogenic methane. Methane is theorised to have been oxidised by chlorine radicals. The authors argue that substantial continuous chlorine radical production is required to explain the observations, which is theorised to be produced by iron photochemistry.

In addition to this, the authors present modelling results that model increased HCHO columns that would follow after methane removal interventions that artificially initiated the processes discussed in the HTHH case.

Data Validity

The data from TROPOMI, VIIRS and MLS is presented principally in Figures 3 and Table 1a & 1b. These data appear to be valid and robust in showing elevated stratospheric HCHO that is coincident with common plume tracers. These HCHO enhancements are expressed in both relative quantities to the tracers and absolute amounts. These calculations and their associated uncertainties seem valid.

The highly enhanced HCHO is a significant result. Enhanced HO₂ is also reported from MLS, but this is only briefly referenced in text (lines 298-300). Given the importance of this to later argument, these HO₂ observations should be presented in either main or supplemental figures.

Discussion

Much of this work is a discussion of what can be derived from these observations with respect to methane and reactive chlorine in the plume. Ozone depletions within this plume had been observed by Evan et al. (2023), Zhu et al. (2023) modelled how it could be attributed to chlorine chemistry. The authors here note that the high HCHO and HO₂ observations do not agree with Zhu et al. (2023)’s outputs for those species used to explain the ozone depletion.

The authors propose that volcanogenic methane, not included in Zhu et al. (2023)’s modelling, is part of the explanation for this. As discussed, it is plausible that HTHH could have emitted sufficient methane to cause this. Unfortunately, as discussed, there are no direct observations of methane enhancement in the plume. An upper bound derived from ACE-FTS profiles is does not preclude emission that would be sufficient to cause the HCHO enhancement.

While Zhu et al. (2023) had modelled an injection of ClO with the eruption, the authors here determine that explaining the persistent elevated HCHO and HO₂, as well as the observed ozone depletions, requires continuous activation of chlorine within the plume.

Bromine chemistry is dismissed as a potential mechanism here, on the basis that observed BrO is too low. There is evidence that the plume from the bromine-rich part of the eruption was separate from this plume that is the focus of this study. I would however still want to see greater discussion of this mechanism before it is dismissed. While the BrO content of this part of the plume is lower, this does not fully evidence that the reactive bromine content of this part of the plume is effectively nil. Is the bromine content of this part of the plume constrained significantly by observations to consider its impacts to be negligible? The reference here to von Glasow (2010) is not adequate, that paper considers principally bromine chemistry of passively degassing plumes within the troposphere, and the field has advanced significantly since 2010. For example, Narivelo et al. (2023) modelled bromine cycling releasing significant reactive chlorine from chloride-containing aerosol. Addressing bromine cycling is necessary to support the statement on line 458 regarding known chlorine production mechanisms.

The authors reach the conclusion that iron photochemistry is the a plausible driving force for their observations. This is an interesting proposition, and, as written, appropriately weighted in terms of how well the evidence supports this, given that several theoretical steps are required to reach this conclusion. Overall the authors make a reasonable case this should be considered. I would however suggest a slight change on line 461-462: “This is the first evidence that iron-chloride chemistry **may be** occurring outside the marine boundary layer.”

Modelling study

The observations and analyses of these are placed after a short modelling discussion.

This model study does not directly replicate the HTHH eruption considered in the main part of the paper, instead considering a theoretical artificial methane-removal effort. Neither the variance in output with differing NO_x concentrations, nor the non-linear relationship between Cl₂/CH₄ oxidation enhancement and HCHO enhancement have significant relationship to the subject of the main body of this manuscript.

I do not see the necessity of this modelling study, and I believe the paper would be better without it. The removal of lines 144-181, 463-465, and 524-540 would not require considerable changes to the rest of the manuscript.

Scope and significance

I find this to be an interesting and informative paper centred around a robust set of observations of the HTHH plume. The record high HCHO observation is a significant observation. The authors make reasonable derivations to produce a plausible explanation for the observed phenomena.

The introduction and aforementioned modelling study however frame this work in the context of artificial methane destruction efforts. I believe this connection is too tenuous, and the paper would be better introduced and considered an observation of natural atmospheric/volcanic phenomena, and any connection to artificial methane destruction could be addressed in terms of a few paragraphs.

If this paper is re-contextualised as primarily an Earth-observation study, it would be a strong paper with notable results. It would however be a question for the editors whether it would be better submitted to a specialised journal.

Minor comments

Units appear to be depicted inconsistently throughout the manuscript, e.g. lines 170-171 switch between unit/unit and unit unit-x format.

Figure 4 needs some refinement – it would be better to combine these series onto one plot. It should also be made clear this is modelled/calculated rather than observed.

The content of figure 5 does not match the caption

Line 421-422, should the g Cl₂ per g photoactive Fe value be “per day”?

Line 443 – this should read “Conclusion”.

The formatting of references 33 and 43 have minor errors.

References used in this review

EVAN, S., BRIOUDE, J., ROSENLOF, K. H., GAO, R.-S., PORTMANN, R. W., ZHU, Y., VOLKAMER, R., LEE, C. F., METZGER, J.-M., LAMY, K., WALTER, P., ALVAREZ, S. L., FLYNN, J. H., ASHER, E., TODT, M., DAVIS, S. M., THORNBERRY, T., VÖMEL, H., WIENHOLD, F. G., STAUFFER, R. M., MILLÁN, L., SANTEE, M. L., FROIDEVAUX, L. & READ, W. G. 2023. Rapid ozone depletion after humidification of the stratosphere by the Hunga Tonga Eruption. *Science*, 382, eadg2551.

NARIVELO, H., HAMER, P. D., MARÉCAL, V., SURL, L., ROBERTS, T., PELLETIER, S., JOSSE, B., GUTH, J., BACLES, M., WARNACH, S., WAGNER, T., CORRADINI, S., SALERNO, G. & GUERRIERI, L. 2023. A regional modelling study of halogen chemistry within a volcanic plume of Mt Etna's Christmas 2018 eruption. *Atmos. Chem. Phys.*, 23, 10533-10561.

VON GLASOW, R. 2010. Atmospheric chemistry in volcanic plumes. *Proceedings of the National Academy of Sciences*, 107, 6594-6599.

ZHU, Y., PORTMANN, R. W., KINNISON, D., TOON, O. B., MILLÁN, L., ZHANG, J., VÖMEL, H., TILMES, S., BARDEEN, C. G., WANG, X., EVAN, S., RANDEL, W. J. & ROSENLOF, K. H. 2023. Stratospheric ozone depletion inside the volcanic plume shortly after the 2022 Hunga Tonga eruption. *Atmos. Chem. Phys.*, 23, 13355-13367.

Reviewer #3

(Remarks to the Author)

This manuscript provides evidence of HCHO production, arguably from methane in the Hunga plume, within days of the eruption based on TROPOMI VCDs. It also provides three simple empirical calculations to estimate the HCHO production, and as a result, the methane removal based on enhancements, and related to other trace gases or aerosol optical depth. The manuscript is interesting and could be a significant contribution (or could be split into multiple papers) but should be revised and resubmitted for the following reasons. First, the manuscript is poorly organized, meaning that numerous passages are repetitive (for instance, two separate discussion sections exist), meandering and confusing. It is not clear how geo-engineering strategies in the marine boundary layer related to methane removal and its monitoring over the ocean warrants discussion in the same paper as observations of the stratospheric HCHO enhancement in the Hunga plume. The authors need to spend more time demonstrating conclusively that the HCHO VCDs are stratospheric based on their collocation with other stratospheric enhancements for a period of several days, and the modeling should be focused on stratospheric conditions. Second, the authors discuss at length how HCHO is evidence of activated chlorine, however, it is not immediately clear that the oxidation of HCHO would be dominated by Cl. The Hunga Tonga plume had an abundance of OH given the large injection of water vapor (Zhu et al., 2022), which is not discussed here. The high OH has been used to explain varying ratios of SO₂/H₂SO₄ in different parts of the plume, with widely varying water vapor mixing ratios (Asher et al., 2023). A third, more minor but still important problem with this manuscript is the correlation of HCHO with sulfate aerosol derived from the EUMETSAT Volcanic Ash RGB 0-255 values and the reliance on MLS data that failed QA. A better way to estimate the sulfate aerosol load would be use to the method in Asher et al., 2023 (e.g., the OMPLS-LP aerosol optical depth and column mass observed from balloon profiles in the early plume) than the EUMETSAT Volcanic Ash RGB 0-255 values, the latter of which is not a quantitative measure. As the authors note, the HCHO enhancement persists for several days and

in some cases the corresponding MLS data did not fail QA – these examples should be featured in the main body of the paper and its figures.

Detailed Comments

Abstract L 21 – 25 The abstract is awkward and vague (confusing). State the sink. Also, are you talking about geoengineering faster atmospheric removal of methane? If so, be more explicit.

HCHO is formed by many compounds – can you clearly explain your argument why it must have been methane that was injected as part of the plume?

L64 – citation for the almost half of methane production from wetlands? Is this poorly constrained?

65-66 Sink still not stated

L110 This sentence is awkward, please rewrite.

Figure 3 Odd to rely so much on MLS data that did not pass QA screening in Figure 3. Could this be swapped with another figure from a later day? Also, what version is the MLS data? This needs to be stated either here or in the methods section.

L124 – While methane emissions happen at the surface atmospheric removal of methane can happen anywhere, even for instance in the stratosphere. Is it possible to separate HCO in the BL, vs free troposphere vs. stratosphere?

L210 – Useful to note that SO₂: sulfate aerosol ratios differ in different parts of the Hunga Tonga plume (Asher et al., 2023 PNAS).

L163 – I, the reader, need more background on the climate intervention – is this easily done and similar to natural dust storms? This is currently <2X the TROPOMI detection limit, so that does not seem ideal.

137– This sentence is awkward, please rewrite.

Modeling takes place in MBL – why is this analogous to Hunga? Why are the box model and CESM not run to simulate conditions in the stratosphere?

Tables 1 and 2: show lat long with degree symbol and E/W, N/S. What is the “confidence level”, 95%? State this or alpha. Also, where is the difference between “incomplete data, N/A, missing data” etc... explained? Table 1b there are two entire blank rows (please remove/correct). Finally, write SO₂ < DL not “SO₂ too low”

What are the experimental conditions in figure 4: pressure temperature etc...? Could you find any TROPOMI overpasses over land where you can also detect CH₄ in the plume? E.g., over Australia? This would be a nice addition to the paper.

L290 – Also noted in Evan et al., 2023.

L294 – Explain why this is surprising

L298 – What version is the HO₂ MLS data and does it pass QA? If not, it should likely not be used without guidance from the MLS team.

L303 – “can only be” replace with “is most likely”

L355-363 Overly negative and at the same time vague – unpack the differences in chemical mechanisms.

L359 – (rate is not observed but inferred based on relationship to HCHO and other variables).

L400 – Check the date – do you mean “Jan. 16” instead of “Feb. 16”?

L443 – Why are there two discussion sections? (previously also L301)

L475 – but not HO₂ (from MLS why not?)

L468 – 481 This entire section belongs in introduction. There is no mention of the Hunga eruption in the introduction which seems odd considering it is central to the paper.

L481 – One of the other most interesting features of the Hunga eruption was the rapid conversion of SO₂ to aerosol (Legras 2022; Asher et al., 2023) which can be explained by elevated OH in the water rich plume...As mentioned above, with an abundance of OH, wouldn't CH₄ be rapidly oxidized even without Cl? If not, please explain.

L548 – Methodologies – you should show equations for clarity (even if they are very simple). Also, Method 2 you say is more accurate but it would seem to introduce new uncertainty to the estimate (based on the uncertainty in other data).

L560- 568 How is the empirical AMF correction derived? We need more details on this or a citation.

References

[1]

Y. Zhu et al., “Perturbations in stratospheric aerosol evolution due to the water-rich plume of the 2022 Hunga-Tonga eruption,” *Commun Earth Environ*, vol. 3, no. 1, p. 248, Oct. 2022, doi: sc.

[1]

E. Asher et al., “Unexpectedly rapid aerosol formation in the Hunga Tonga plume,” *Proc. Natl. Acad. Sci. U.S.A.*, vol. 120, no. 46, p. e2219547120, Nov. 2023, doi: 10.1073/pnas.2219547120.

Version 1:

Reviewer comments:

Reviewer #1

(Remarks to the Author)

I have reviewed the authors' rebuttal and revised manuscript. I am satisfied the way my comments have been addressed. I have no further concerns and I recommend the manuscript for publication.

Reviewer #2

(Remarks to the Author)

This is my second view of the manuscript, having previously reviewed an earlier draft. My major comments concerned:

*a modelling section that has now been removed

* the discussion of the bromine chemistry which has been expanded.

The authors have provided detailed responses to my and the other reviewers' comments.

I am satisfied that all of my comments have been adequately addressed. As long as the other reviewers are also satisfied I am happy to recommend the paper for publication in its current form.

I would suggest some minor formatting improvements to Figure 2 to bring this to publication quality:

*Remove unnecessary decimal places from the values

*Format j_{HCHO} with actual subscript text rather than using an underscore

*Replacing "\s" with "/s" or using a superscript "-1".

Reviewer #3

(Remarks to the Author)

Thank you for the thorough revision. I now find this article suitable for publication and have no further comments.

Our responses to the reviewer comments are below in blue.

New text is written in red.

REVIEWER COMMENTS

Reviewer #1 (Remarks to the Author):

General assessment

The manuscript presents compelling observational evidence of extremely elevated HCHO in the Hunga Tonga eruption plume, and the methodological framework is sound. I agree with the authors' interpretation that heterogeneous activation of chlorine on sulfate aerosols alone cannot account for the inferred chlorine production rates. The proposed contribution from iron photochemistry acting on sulfate-coated ash particles is plausible and consistent with recent advances in our understanding of post-HTHH aerosol composition. The results merit publication after several clarifications and a moderation of some claims.

Thank you very much for your review.

Major comments

1) Lines 461–462 state that this work provides “the first evidence for iron chloride chemistry occurring outside the marine boundary layer.”

I do not think the current dataset supports this level of certainty. The mechanism is plausible and consistent with the observed HCHO and ClO enhancements, but the attribution remains indirect and the mechanism is not uniquely constrained by the observations presented. A full chemistry–transport evaluation is essential before such a statement can be supported.”

I recommend softening the claim. For example:

“...our analysis suggests that iron–chloride photochemistry may be active in the stratosphere, but confirmation will require dedicated modeling and laboratory studies.”

As you suggested, we changed it to: “Our analysis suggests that iron–chloride photochemistry may be active in the stratosphere, but confirmation will require dedicated modeling and laboratory studies.”

2) The abstract currently states that “known chlorine production mechanisms cannot explain the high chlorine production rate.”

This should be nuanced. The HTHH eruption is a unique case with exceptional water vapor and aerosol loadings in the stratosphere. Existing parameterizations of heterogeneous chlorine activation were developed for very different regimes and likely require revision before firm conclusions can be drawn. The mechanisms may still be valid, but their parameterizations at extremely high H₂O and aerosol surface area conditions are likely outside their domain of applicability.

I recommend reformulating this to:

“Under the extreme conditions of the HTHH plume, current parameterizations of chlorine activation appear insufficient to account for the inferred production rate.”

To acknowledge the uncertainty due to the extreme conditions of the HHTH plume, we will nuance the sentence to: “We show that currently known chlorine production mechanisms appear insufficient to explain such a high chlorine production rate”.

You are correct that the extreme conditions of the HTHH plume change heterogeneous chlorine activation, but we are making this statement based on references that have adapted the parameterizations to these conditions. Most relevant is the study by Evan2023, who calculated the strong dependence of the reactive uptake probability γ on H₂O concentration and temperature, which decreases substantially during the days covered by our observations. As argued in our manuscript, this mechanism cannot explain our observation that $\Delta\text{HCHO}/\Delta\text{AOD}$ enhancement ratios remained stable after Jan 17 and that by Jan-25 the H₂O enhancement becomes partly separated from the ClO/HO₂ enhancement.

Also note that we are writing ‘known chlorine production *mechanisms*’, instead of ‘parameterizations’, because our argumentation is also directed at the mechanisms themselves.

1. Evan S, Brioude J, Rosenlof KH, Gao RS, Portmann RW, Zhu Y, Volkamer R, Lee CF, Metzger JM, Lamy K, Walter P, Alvarez SL, Flynn JH, Asher E, Todt M, Davis SM, Thornberry T, Vömel H, Wienhold FG, Stauffer RM, Millán L, Santee ML, Froidevaux L, Read WG. Rapid ozone depletion after humidification of the stratosphere by the Hunga Tonga Eruption. *Science*. 2023 Oct 20;382(6668):eadg2551. doi: 10.1126/science.adg2551.

3) In the abstract (line 37), the authors state that iron–chloride photochemistry is a plausible mechanism.

Based on the presented correlations and scaling arguments, I agree that the mechanism is plausible. I would, however, avoid calling this “evidence” in the absence of a full chemical box or 3-D model evaluation. The authors may instead say:

“Our analysis supports iron–chloride photochemistry as a strong candidate mechanism.”

Your comment is similar to your comment 1) on the conclusion section. We have made a similar change to the abstract, changing “can be active” into “**may be active**”.

4) To substantiate the proposed mechanism, a targeted chemical modeling experiment would be required (e.g., WACCM, CESM, or a plume-resolving model including iron photochemistry). The current observational correlations cannot exclude competing or complementary pathways.

I suggest adding a statement acknowledging this limitation and indicating that model evaluation is needed to test the hypothesis.

We agree and have added your suggestion to the conclusion section. Such a model should also include the methane injection, because this substantially changes the chlorine chemistry as well:

“...but confirmation will require dedicated modeling and laboratory studies (e.g., a global or plume-resolving model including iron photochemistry and methane injection).”

Minor comments

1. the authors might want to add a short paragraph in the method section that explains the difference between VCD, SCD and the assumptions to estimate a concentration. That will help the reader unfamiliar with satellite retrievals

We added the following to the methods:

“Vertical column density (VCD) represents the amount of a trace gas integrated straight down through the atmosphere, while slant column density (SCD) is the amount measured along the actual, typically angled, light path through the atmosphere. Because the slanted path is longer than the vertical path, the SCD is usually larger than the VCD for the same atmospheric state. The air mass factor (AMF) relates the two by $VCD = SCD / AMF$. The value of the AMF depends on the length of the light path, the vertical distribution of absorbing trace gases in the atmosphere, the reflectivity (albedo) of the earth's surface, the presence of aerosols and clouds, etc.”

2. The authors should also explain a bit further how the HTHH eruption has affected the MLS and other satellites retrievals. It will help to better understand the discussions behind “quality screening”.

For a discussion of quality screening we refer to relevant papers for more information.

Regarding quality screening for MLS data, it is explained in the results section that: “In the initial days most of the MLS data did not pass the standard Quality Screening (MLS), which was attributed to extremely enhanced H₂O at very high altitudes (26;Millán2022).”

In the case of ACE-FTS the aerosol cloud blocks visibility of the sun, leading to pointing errors in the instrument (the instrument needs to be pointed directly at the sun, but due to the obstruction by the aerosol cloud the pointing becomes misaligned). This is explained in the discussion section:

“(32;Bernath2023). Unfortunately, due to pointing jumps caused by high aerosol extinction the CH₄ observations are invalid for this day. “

3. It would also be interesting to briefly comment on whether similar iron–chloride photochemical pathways might have been relevant for past eruptions such as Pinatubo, and whether the much lower stratospheric water vapor during that event would have limited or enhanced such chemistry.

We added the following to the discussion:

“The iron-chloride photochemistry mechanism may not be as significant in other volcanic eruptions, because the HHTH eruption provided unique conditions favorable to iron-chloride photochemistry. This includes the exceptionally large sea-water injection that also injected a large amount of sea-salt needed for the mechanism. At the same time the SO₂ emission was relatively modest (reducing potential inhibition by sulphate). “

Reviewer #2 (Remarks to the Author):

Reviewer report on: Satellite quantification of enhanced methane oxidation applied to the stratospheric plume following Hunga Tonga-Hunga Ha’apai eruption

Key results

In this manuscript, the authors report an analysis of chemistry of the plume of the Hunga Tonga-Hunga Ha’apai (HTHH) eruption of January 2016, which was emitted into the stratosphere. Using data from

satellite observations, principally TROPOMI, high amounts of HCHO are shown to be present in two clouds this stratospheric plume. The peak HCHO concentration is determined to be the highest ever observed within the stratosphere.

This HCHO is found to be persistent within the plume – despite HCHO’s expected short lifetime to photolytic destruction. The authors argue is evidence of continual HCHO formation from volcanogenic methane. Methane is theorised to have been oxidised by chlorine radicals. The authors argue that substantial continuous chlorine radical production is required to explain the observations, which is theorised to be produced by iron photochemistry.

In addition to this, the authors present modelling results that model increased HCHO columns that would follow after methane removal interventions that artificially initiated the processes discussed in the HTHH case.

Thank you very much for your review.

Data Validity

The data from TROPOMI, VIIRS and MLS is presented principally in Figures 3 and Table 1a & 1b. These data appear to be valid and robust in showing elevated stratospheric HCHO that is coincident with common plume tracers. These HCHO enhancements are expressed in both relative quantities to the tracers and absolute amounts. These calculations and their associated uncertainties seem valid. The highly enhanced HCHO is a significant result. Enhanced HO₂ is also reported from MLS, but this is only briefly referenced in text (lines 298-300). Given the importance of this to later argument, these HO₂ observations should be presented in either main or supplemental figures.

In response to your comment, we have added a new figure that shows the MLS data and also some additional HCHO/AOD data for a selection of observations, including the HO₂ data for c2b_21 and c2b_25 that is used in the discussion of the manuscript. We will also add the MLS data to the data-access files.

Figure: Correlation between HCHO, AOD and MLS observations for clouds C2_17 (top), C2b_21 (middle) and C2b_25 (bottom). While initially MLS H₂O enhancement is co-located with HCHO/AOD/HO₂ enhancement, by Jan-25 the H₂O enhancement becomes separated from the other enhancements. MLS data with an X did not pass quality screening. We did not apply quality screening to MLS H₂O, but the data can be trusted to show the location of the H₂O enhancement (26;Millán2022).

Discussion

Much of this work is a discussion of what can be derived from these observations with respect to methane and reactive chlorine in the plume. Ozone depletions within this plume had been observed by Evan et al. (2023). Zhu et al. (2023) modelled how it could be attributed to chlorine chemistry. The authors here note that the high HCHO and HO₂ observations do not agree with Zhu et al. (2023)'s outputs for those species used to explain the ozone depletion.

The authors propose that volcanogenic methane, not included in Zhu et al. (2023)'s modelling, is part of the explanation for this. As discussed, it is plausible that HTHH could have emitted sufficient methane to cause this. Unfortunately, as discussed, there are no direct observations of methane enhancement in the plume. An upper bound derived from ACE-FTS profiles is does not preclude emission that would be sufficient to cause the HCHO enhancement.

While Zhu et al. (2023) had modelled an injection of ClO with the eruption, the authors here determine that explaining the persistent elevated HCHO and HO₂, as well as the observed ozone depletions, requires continuous activation of chlorine within the plume.

Bromine chemistry is dismissed as a potential mechanism here, on the basis that observed BrO is too low. There is evidence that the plume from the bromine-rich part of the eruption was separate from this plume that is the focus of this study. I would however still want to see greater discussion of this mechanism before it is dismissed. While the BrO content of this part of the plume is lower, this does not fully evidence that the reactive bromine content of this part of the plume is effectively nil. Is the bromine

content of this part of the plume constrained significantly by observations to consider its impacts to be negligible? The reference here to von Glasow (2010) is not adequate, that paper considers principally bromine chemistry of passively degassing plumes within the troposphere, and the field has advanced significantly since 2010. For example, Narivelo et al. (2023) modelled bromine cycling releasing significant reactive chlorine from chloride-containing aerosol. Addressing bromine cycling is necessary to support the statement on line 458 regarding known chlorine production mechanisms.

Thank you for pointing out the need for a further discussion on bromine chemistry. We have added substantial extra text and analysis (see below).

We will add also your suggested reference to Narivelo et al. (2023), which is indeed a helpful in-depth discussion on the bromine chemistry in volcano plumes. Narivelo also points out another important point, which is that the bromine catalytic cycling is constrained by $\text{Br} + \text{HCHO}$ to form HBr . In our specific case, HCHO is extremely enhanced, which will cause the bromine speciation in the HHTH plume to shift more towards HBr , possibly explaining the relatively low observed BrO compared to SO_2 in the HHTH plume.

Added $\Delta\text{HCHO}/\Delta\text{BrO}$ enhancements to Table 1a, and moved the location data to the supplemental information.

Table 1a. Overview of HCHO enhancement ratios that could be confidently assessed.

Observation ***	$\Delta\text{HCHO}/\Delta\text{SO}_2$ (mmol/mol) *	$\Delta\text{HCHO}/\Delta\text{SA}$ (molec/cm ² per green value) **	$\Delta\text{HCHO}/\Delta\text{AOD}$ (molec/cm ² per AOD) *	$\Delta\text{HCHO}/\Delta\text{BrO}$ (mol/mol) *
C1_16	$4.5 \pm 11\%$ ($r=0.76$)	$2.0\text{E}+13 \pm 12\%$ ($r=0.81$)	$1.3\text{E}+15 \pm 08\%$ ($r=0.86$)	$41.1 \pm 17\%$ ($r=0.62$)
C1a_19	$40.0 \pm 07\%$ ($r=0.57$)	missing data	missing data	$12.1 \pm 29\%$ ($r=0.17$)
C1b_20	low SO_2	$6.8\text{E}+12 \pm 22\%$ ($r=0.41$)	$4.0\text{E}+14 \pm 12\%$ ($r=0.54$)	$19.0 \pm 22\%$ ($r=0.27$)
C2_16	$1.0 \pm 06\%$ ($r=0.80$)	$1.0\text{E}+13 \pm 08\%$ ($r=0.85$)	no data	$5.5 \pm 10\%$ ($r=0.65$)
C2_17	$1.1 \pm 09\%$ ($r=0.59$)	$7.3\text{E}+12 \pm 12\%$ ($r=0.61$)	$2.2\text{E}+14 \pm 16\%$ ($r=0.48$)	$6.4 \pm 08\%$ ($r=0.62$)
C2a_20	$2.2 \pm 20\%$ ($r=0.42$)	$2.0\text{E}+12 \pm 37\%$ ($r=0.32$)	$1.7\text{E}+14 \pm 27\%$ ($r=0.38$)	$6.5 \pm 23\%$ ($r=0.37$)
C2b_21	too low SO_2	$5.3\text{E}+12 \pm 14\%$ ($r=0.54$)	$3.7\text{E}+14 \pm 14\%$ ($r=0.53$)	$21.8 \pm 11\%$ ($r=0.51$)
C2b_22	too low SO_2	$3.7\text{E}+12 \pm 29\%$ ($r=0.33$)	$1.9\text{E}+14 \pm 25\%$ ($r=0.35$)	$6.4 \pm 32\%$ ($r=0.20$)
C2b_25	too low SO_2	$7.3\text{E}+12 \pm 27\%$ ($r=0.31$)	$2.6\text{E}+14 \pm 15\%$ ($r=0.39$)	$10.1 \pm 20\%$ ($r=0.24$)
C3_23	N/A ($r=-0.04$)	missing data	N/A ($r=-0.07$)	N/A ($r=-0.36$)

Added text:

“We analyzed the correlation between TROPOMI BrO and HCHO enhancement (see Table 1a), and found a generally linear correlation with $\Delta\text{HCHO}/\Delta\text{BrO}$ varying from 6 to 40 mol/mol. The correlation between BrO and HCHO is generally weaker compared to the AOD/HCHO correlation, with poor

BrO/HCHO correlation for cloud C1. For cloud C2, the $\Delta\text{HCHO}/\Delta\text{BrO}$ enhancement ratio remained stable between Jan-16 and Jan-25. “

“In a typical volcanic plume, bromine chemistry is a key mechanism for chlorine activation (36;vonGlasgow2010, 54;Narivelo2024), in which a catalytic cycle involving Br activates Cl, while depleting ozone. According to Zhu, bromine chemistry cannot explain the Cl production in the HHTH plume, because it would imply a much stronger ozone depletion than was observed (25;Zhu2023). In addition, BrO was observed at a different moment during the HTHH eruptions and reached a lower altitude of 8-15 km, where a different wind direction spread the BrO in the opposite southeastward direction compared to the plume that we investigate here (37;Li2023). Bromine catalytic cycling is constrained by the Br + HCHO reaction that forms HBr (54;Narivelo2024). This shifts bromine speciation towards HBr within our observed strong HCHO enhancements, possibly explaining the relatively low observed BrO compared to SO₂ in the high-altitude stratospheric HHTH plume that we investigate, and limiting bromine chemistry as Cl source.

Despite these arguments, we still observe a modest BrO enhancement and it is correlated with HCHO (see Table 1a, and Figures S2 – S19). We calculated the maximum rate of Cl production through bromine chemistry by calculating the rate of formation and reactive uptake of HOBr using observed values for BrO, HO₂ and aerosol surface area for cloud C2b_21 (see Supplemental Information Text). We find that the maximum Cl production is $1.5 \times 10^4 \text{ cm}^{-3}\text{s}^{-1}$, while our observed value is an order of magnitude larger at $3 \times 10^5 \text{ cm}^{-3}\text{s}^{-1}$. We therefore conclude that Br activation of Cl cannot explain the majority of our observed Cl production. “

Added Bromine observations to SI figures, for example C2b_21 below:

Added to conclusion:

“Bromine emissions might have been higher than current estimates, because HCHO shifts speciation away from BrO towards HBr, masking much of the emission. “

Added to Supplemental Information:

Calculation of primary Cl production through HOBr reactive uptake

For Jan-21 we observed a peak BrO enhancement of $2.5 \times 10^{13} \text{ molec/cm}^2$, which corresponds with an estimated average concentration of $1.3 \times 10^9 \text{ cm}^{-3}$ (0.2 ppb). Combined with the peak MLS observations of HO₂ (0.7 ppb) the peak rate of HOBr formation for Jan-21 is $3.2 \times 10^6 \text{ cm}^{-3}\text{s}^{-1}$.

The reactive uptake coefficient for HOBr is strongly dependent on temperature. Based on MLS observations the temperature of C1b_21 was 220 K (26; Millán2022) (see figure S11). At this temperature the maximum possible reaction probability $\gamma = 1 \times 10^{-3}$ based on (Zhang2024).

The fraction of HOBr production that leads to chlorine amplification depends on the competition between HOBr photolysis rate ($3.3 \times 10^{-3} \text{ s}^{-1}$) and reactive uptake rate (first order rate of $1.6 \times 10^{-5} \text{ s}^{-1}$ based on the maximum observed aerosol surface area density of $2.9 \times 10^{-6} \text{ cm}^2 \text{ cm}^{-3}$, (27;Evan2023), $\gamma = 1 \times 10^{-3}$ and thermal velocity $v = 219 \text{ m/s}$. This means approximately 0.5% of HOBr production may lead to

amplification. This means that the maximum Cl production through this mechanism is $1.5 \times 10^4 \text{ cm}^{-3}\text{s}^{-1}$, while our observed value is an order of magnitude larger at $3 \times 10^5 \text{ cm}^{-3}\text{s}^{-1}$.

In addition to this, the observed correlation between HCHO and BrO was linear, while a non-linear correlation is expected if the mechanism depends on the combination of BrO, HO₂ and aerosols. For example, high BrO is associated with high aerosol optical depth and also associated with high HO₂, due to which the HCHO:BrO enhancement ratio would increase for higher BrO enhancements (which is not what we observed).

We therefore conclude that the observed correlation between HCHO and BrO enhancement is due to Br activation by Cl that is produced through another mechanism (instead of Br activating the Cl).

The authors reach the conclusion that iron photochemistry is the a plausible driving force for their observations. This is an interesting proposition, and, as written, appropriately weighted in terms of how well the evidence supports this, given that several theoretical steps are required to reach this conclusion. Overall the authors make a reasonable case this should be considered. I would however suggest a slight change on line 461-462: "This is the first evidence that iron-chloride chemistry ****may be**** occurring outside the marine boundary layer."

As you and the other reviewer suggested, we changed it to: "Our analysis suggests that iron–chloride photochemistry may be active in the stratosphere, but confirmation will require dedicated modeling and laboratory studies."

*****Modelling study*****

The observations and analyses of these are placed after a short modelling discussion. This model study does not directly replicate the HTHH eruption considered in the main part of the paper, instead considering a theoretical artificial methane-removal effort. Neither the variance in output with differing NO_x concentrations, nor the non-linear relationship between Cl₂/CH₄ oxidation enhancement and HCHO enhancement have significant relationship to the subject of the main body of this manuscript. I do not see the necessity of this modelling study, and I believe the paper would be better without it. The removal of lines 144-181, 463-465, and 524-540 would not require considerable changes to the rest of the manuscript.

According to your suggestion, we have removed the modelling of the theoretical artificial methane-removal effort.

*****Scope and significance*****

I find this to be an interesting and informative paper centred around a robust set of observations of the HTHH plume. The record high HCHO observation is a significant observation. The authors make reasonable derivations to produce a plausible explanation for the observed phenomena.

The introduction and aforementioned modelling study however frame this work in the context of artificial methane destruction efforts. I believe this connection is too tenuous, and the paper would be better introduced and considered an observation of natural atmospheric/volcanic phenomena, and any connection to artificial methane destruction could be addressed in terms of a few paragraphs.

If this paper is re-contextualised as primarily an Earth-observation study, it would be a strong paper with

notable results. It would however be a question for the editors whether it would be better submitted to a specialised journal.

According to your suggestion, we have shortened the artificial methane removal section in the introduction, added HHTH to the introduction, and removed the modelling. The artificial removal discussion is now reduced to the following paragraph in the conclusion:

“Meidan et al. (21;Meidan2024) modelled local emission of iron for atmospheric methane removal over the ocean and found 25 Gg Cl per hour removed 3.1 Gg CH₄ per hour, reducing global radiative forcing by 0.04 W m⁻² within 10 years. This removal amount is much higher than our observed HHTH removal of 75±18 Mg CH₄ per hour at midday, which was clearly detectable. Therefore, the sensitivity of our methodology can be sufficient for quantification in hypothetical future enhanced atmospheric methane oxidation approaches to help address future global warming. “

We believe these changes address your comments, while also keeping the manuscript attractive for a broad audience, including from earth-observation interest in HHTH, to the new emerging field of atmospheric methane removal that can benefit from our results in future research.

Minor comments

Units appear to be depicted inconsistently throughout the manuscript, e.g. lines 170-171 switch between unit/unit and unit unit-x format.

We are consistently using unit/unit for column densities and and unit-x for concentrations and rates. This is done to help the reader more easily distinguish between these different types of units. This manuscript will be read by researchers that are used to working with column densities as measured by satellite observation, and will also be read by atmospheric chemists that are more accustomed to work with concentrations and rates. To make this clear, the manuscript included a sentence already, but we will move this to a more predominant location now (the start of the results section):

“Note that we will report column density in units of molec/cm² (with /), concentrations in cm⁻³, and rates in cm⁻³ s⁻¹, to clearly distinguish between them. “

Figure 4 needs some refinement – it would be better to combine these series onto one plot. It should also be made clear this is modelled/calculated rather than observed.

Updated (see below). We also updated figure 5 in the same way.

Figure 4. HCHO in-plume photolysis rate as a function of time (solid line), starting from the moment of the eruption (based on NCAR TUV calculator). At the TROPOMI overpass time (13:30, marked with a vertical line), a photolysis rate of $1.14 \cdot 10^{-4} \text{ s}^{-1}$ corresponds with a HCHO lifetime of 2.5 hours. The dashed line shows the percentage of HCHO remaining if it were injected by the eruption without in-plume formation, showing 95% HCHO reduction for the jan-16 TROPOMI overpass, and 99.95% reduction for the second overpass.

The content of figure 5 does not match the caption

Thanks. We deleted '~~The blue line shows the high estimate and orange shows the low estimate~~' (it shows the central estimate now and the error margin is in the main text).

Line 421-422, should the g Cl₂ per g photoactive Fe value be "per day"?

You are correct, we added '~~per day~~' in the text.

Line 443 – this should read "Conclusion".

Thanks, we have updated this.

The formatting of references 33 and 43 have minor errors.

Thanks, we have updated this.

References used in this review

EVAN, S., BRIOUDE, J., ROSENLOF, K. H., GAO, R.-S., PORTMANN, R. W., ZHU, Y., VOLKAMER, R., LEE, C. F., METZGER, J.-M., LAMY, K., WALTER, P., ALVAREZ, S. L., FLYNN, J. H., ASHER, E., TODT, M., DAVIS, S. M., THORBERRY, T., VÖMEL, H., WIENHOLD, F. G., STAUFFER, R. M., MILLÁN, L., SANTEE, M. L., FROIDEVAUX, L. & READ, W. G. 2023. Rapid ozone depletion after humidification of the stratosphere by the Hunga Tonga Eruption. *Science*, 382, eadg2551.

NARIVELLO, H., HAMER, P. D., MARÉCAL, V., SURL, L., ROBERTS, T., PELLETIER, S., JOSSE, B., GUTH, J., BACLES, M., WARNACH, S., WAGNER, T., CORRADINI, S., SALERNO, G. & GUERRIERI, L. 2023. A regional modelling study of halogen chemistry within a volcanic plume of Mt Etna's Christmas 2018 eruption. *Atmos. Chem. Phys.*, 23, 10533-10561.

VON GLASOW, R. 2010. Atmospheric chemistry in volcanic plumes. Proceedings of the National Academy of Sciences, 107, 6594-6599.

ZHU, Y., PORTMANN, R. W., KINNISON, D., TOON, O. B., MILLÁN, L., ZHANG, J., VÖMEL, H., TILMES, S., BARDEEN, C. G., WANG, X., EVAN, S., RANDEL, W. J. & ROSENLOF, K. H. 2023. Stratospheric ozone depletion inside the volcanic plume shortly after the 2022 Hunga Tonga eruption. Atmos. Chem. Phys., 23, 13355-13367.

Reviewer #3 (Remarks to the Author):

This manuscript provides evidence of HCHO production, arguably from methane in the Hunga plume, within days of the eruption based on TROPOMI VCDs. It also provides three simple empirical calculations to estimate the HCHO production, and as a result, the methane removal based on enhancements, and related to other trace gases or aerosol optical depth. The manuscript is interesting and could be a significant contribution (or could be split into multiple papers) but should be revised and resubmitted for the following reasons. First, the manuscript is poorly organized, meaning that numerous passages are repetitive (for instance, two separate discussion sections exist), meandering and confusing.

We apologize for the confusion. The second 'discussion' section is meant to be a 'conclusion' section, and this has now been adapted.

It is not clear how geo-engineering strategies in the marine boundary layer related to methane removal and its monitoring over the ocean warrants discussion in the same paper as observations of the stratospheric HCHO enhancement in the Hunga plume.

According to your suggestion below, we have removed the modeling of the theoretical artificial methane-removal effort in the marine boundary layer (lines 169-181, 426-430, 463-465). We agree it is better to publish this in another paper.

The authors need to spend more time demonstrating conclusively that the HCHO VCDs are stratospheric based on their colocation with other stratospheric enhancements for a period of several days, and the modeling should be focused on stratospheric conditions.

In response to your comment, we have added a new figure that shows the MLS data and also some additional HCHO/AOD data for a selection of observations, including the HO₂ data for c2b_21 and c2b_25 that is used in the discussion of the manuscript (see below in our response to your comment on MLS).

We have also elaborated in other sections on the stratospheric chemistry conditions, including new calculations to respond to some of your comments below about OH and SO₂ oxidation and the other reviewer comments on Bromine.

Second, the authors discuss at length how HCHO is evidence of activated chlorine, however, it is not immediately clear that the oxidation of HCHO would be dominated by Cl. The Hunga Tonga plume had an abundance of OH given the large injection of water vapor (Zhu et al., 2022), which is not discussed here.

The high OH has been used to explain varying ratios of SO₂/ H₂SO₄ in different parts of the plume, with widely varying water vapor mixing ratios (Asher et al., 2023).

To address your comment, we have now added additional discussion and calculations that constrain OH and Cl based on the observed linear relationship between HCHO and other species in the HHTH cloud. We used this to also constrain the minimum required CH₄ emission by the eruption. Thank you for this comment, because this extra analysis has strengthened the conclusions.

The added text to the discussion section is:

“We have observed a linear correlation between HCHO and aerosols (SA and AOD), which suggests that HCHO lifetime at midday is mainly driven by photolysis (see Methods equation 1). If HCHO lifetime were limited by OH or Cl produced by the aerosols, then the correlation would not be linear (see methods equation 2). To be consistent with this, HCHO loss to OH and Cl should be below 50%, resulting in a maximum concentration of $1 \times 10^7 \text{ cm}^{-3}$ for OH and $2 \times 10^6 \text{ cm}^{-3}$ for Cl, based on reaction rates (see Table S1 and Fig. S1). “

And:

“We calculated the minimum required methane elevation, by combining the observed methane oxidation rate with the maximum possible OH and Cl concentrations, and with the known reaction rates for methane oxidation by Cl and OH (see table S1). On Jan-16 the area of the HCHO enhancement is approximately $7.2 \times 10^6 \text{ km}^2$ and the thickness is 2 km (23;Legras2022). This yields an average CH₄ oxidation rate of $5.5 \times 10^6 \text{ cm}^3 \text{ s}^{-1}$. If 100% of this is due to a maximum OH enhancement of $1 \times 10^7 \text{ cm}^{-3}$, the CH₄ concentration was at least 95 ppm compared to a background value of 1 ppm (an enhancement of at least 2300 Gg CH₄). If instead 90% is due to a maximum Cl enhancement of $2 \times 10^6 \text{ cm}^{-3}$, the CH₄ concentration was at least 14 ppm (330 Gg CH₄).”

And following the results of the ACE-FTS analysis:

“This rules out the possibility that the HCHO enhancement is due to OH (it would mean at least 2300 Gg CH₄ emission that would be clearly visible on ACE-FTS). However, the emission of 330 Gg CH₄ due to Cl enhancement is realistic and would indeed not have been detectable with ACE-FTS by Feb 2022.

Thus, our observed HCHO enhancement is due to an increase in Cl, combined with an average methane concentration enhancement of at least 14 ppm in the Jan-16 volcanic cloud. “

Added figure S1:

Fig. S1. Comparison of HCHO photolysis rate with the first order loss rate for varying [OH] and [Cl], showing that HCHO loss to OH and Cl is below 50% for a maximum concentration of $1 \times 10^7 \text{ cm}^{-3}$ for OH and $2 \times 10^6 \text{ cm}^{-3}$ for Cl.

A third, more minor but still important problem with this manuscript is the correlation of HCHO with sulfate aerosol derived from the EUMETSAT Volcanic Ash RGB 0-255 values and the reliance on MLS data that failed QA. A better way to estimate the sulfate aerosol load would be use to the method in Asher et al., 2023 (e.g., the OMPLS-LP aerosol optical depth and column mass observed from balloon profiles in the early plume) than the EUMETSAT Volcanic Ash RGB 0-255 values, the latter of which is not a quantitative measure.

As mentioned, the MLS data that we used for our conclusions passed QA.

In addition, for our calculations of Cl production through HOBr and HOCl uptake, we used aerosol surface area data reported by Evan2023 from the same balloon measurements as reported by Asher (so we already used the balloon data you refer to).

We considered to use OMPS-LP, but it was not suitable for our analysis due to a lack of spatial mapping (there are large gaps between adjacent profiles). Instead, VIIRS AOD provides full mapping that we could compare with the spatial mapping of HCHO.

We note that EUMETSAT Volcanic Ash RGB is widely used in studies of the HHTH plume to show the location of sulfate aerosol, and we note that it is also used by Asher (see their Methods). EUMETSAT Volcanic Ash RGB is specifically designed for detecting volcanic ash (exploiting its characteristic 10–12 μm spectral signature) and therefore complements the other data that we analyze. We agree that RGB is a quantitative measure, but it is still suitable for finding correlations between the location of the HCHO enhancement and sulphate aerosol (which is how we use it in our manuscript).

As the authors note, the HCHO enhancement persists for several days and in some cases the corresponding MLS data did not fail QA – these examples should be featured in the main body of the paper and its figures.

Our conclusions are based on MLS observations on Jan-21, and all of these MLS observations passed quality check. The only MLS observations that fail QA for this day (and other days) are H₂O MLS measurements, but it was shown by Millán that these observations can be used to determine the location of the plume (which is also how we use the H₂O observations in our conclusions).

To address your comment on the MLS observations we have added an additional figure with MLS data:

Figure: Correlation between HCHO, AOD and MLS observations for clouds C2_17 (top), C2b_21 (middle) and C2b_25 (bottom). While initially MLS H₂O enhancement is co-located with HCHO/AOD/HO₂ enhancement, by Jan-25 the H₂O enhancement becomes separated from the other enhancements. MLS data with an X did not pass quality screening. We did not apply quality screening to MLS H₂O, but the data can be trusted to show the location of the H₂O enhancement (26;Millán2022).

Detailed Comments

Abstract L 21 – 25 The abstract is awkward and vague (confusing). State the sink. Also, are you talking about geoeengineering faster atmospheric removal of methane? If so, be more explicit.

Clarified to 'Atmospheric sink', and changed to 'a new field of enhanced atmospheric removal is emerging that'.

Note that this field also includes local emission mitigation approaches that don't fit to the definition of geoeengineering. In the main text we refer to a report by the National Academy of Science about the field.

HCHO is formed by many compounds – can you clearly explain your argument why it must have been methane that was injected as part of the plume?

This is addressed in the discussion of the manuscript:

“Our observed HCHO production is so high that the majority of HCHO can only be produced by CH₄ oxidation (the main precursor for HCHO in the stratosphere), and not from non-methane VOCs (NMVOCs) emitted by the volcano. NMVOCs are known to be emitted by volcanoes, but only at trace concentrations that are at least an order of magnitude less than methane (33;Tassi2009). In addition, the seawater concentrations of DMS (1-10 nM) (34;Zhou2024) and dissolved organic carbon (maximum 100

uM) (35;Tiantian2022) are too low to cause a substantial injection of carbon through the 146 Tg stratospheric H₂O injection.”

L64 – citation for the almost half of methane production from wetlands? Is this poorly constrained?

The reference is (3,Saunois2024), now added to the sentence, and our text states ‘natural sources such as wetlands’, thus we say that half of methane is from natural sources (not only wetlands, although wetlands is the majority).

65-66 Sink still not stated

Clarified to ‘Atmospheric sink’

L110 This sentence is awkward, please rewrite.

“The report identified measurement, reporting, monitoring and verification as a key challenge, and concludes that we currently lack tools for methane removal quantification.”

Figure 3 Odd to rely so much on MLS data that did not pass QA screening in Figure 3. Could this be swapped with another figure from a later day? Also, what version is the MLS data? This needs to be stated either here or in the methods section.

We used MLS v5, added to the text now: “with coincidental simultaneous MLS v5 observations”

We also added an additional figure with MLS data for other days (see above responding to your other comment).

L124 – While methane emissions happen at the surface atmospheric removal of methane can happen anywhere, even for instance in the stratosphere. Is it possible to separate HCO in the BL, vs free troposphere vs. stratosphere?

This is not possible with TROPOMI. Also, under normal circumstances HCHO is mainly found in the troposphere. The high stratospheric HCHO observed in our study is highly unusual.

L210 – Useful to note that SO₂: sulfate aerosol ratios differ in different parts of the Hunga Tonga plume (Asher et al., 2023 PNAS).

We added: “We note that previous observations also found that SO₂:SA ratios differ in different parts of the HHTH plume (Asher2023).”

L163 – I, the reader, need more background on the climate intervention – is this easily done and similar to natural dust storms? This is currently <2X the TROPOMI detection limit, so that does not seem ideal.

This part has now been deleted from the manuscript, according to your suggestion above.

137– This sentence is awkward, please rewrite.

Rephrased to: “To demonstrate proof-of-concept, we will apply the method to the Hunga Tonga-Hunga Ha’apai (HHTH) volcanic eruption for which chlorine activation was observed. To the best of our knowledge, this the first analysis of HCHO observations following a volcanic eruption.”

Modeling takes place in MBL – why is this analogous to Hunga? Why are the box model and CESM not run to simulate conditions in the stratosphere?

We had included box modeling to show the general trends as they would be seen in the troposphere, to show the feasibility of the methodology for quantification of artificially enhanced methane oxidation (which would occur in the troposphere). Modeling of the HHTH plume is beyond the scope of this paper (the modeling results that we presented are re-analysis of previous modelling work).

Indeed, there is limited direct relevance between the HHTH eruption and these modeling results, which is why they will be removed (as you and other reviewers suggested).

Tables 1 and 2: show lat long with degree symbol and E/W, N/S.

Done

What is the “confidence level”, 95%? State this or alpha.

Added below the table: “Showing \pm standard error”

Also, where is the difference between “incomplete data, N/A, missing data” etc... explained?

Replaced “N/A” with “no correlation”, and clarified the other ones as ‘invalid data’. For example, data may be incomplete or due to cloud interference, or may not be available at all (missing) due to a missing overpass. We’ve changed it to ‘invalid data’ without being specific with terms like missing or incomplete.

For table 1b, column “Total HCHO enhancement above baseline (mol)*” we have put ‘invalid baseline’, because we could not determine the baseline in the fraction of the cloud.

Table 1b there are two entire blank rows (please remove/correct).

These blank rows are there to distinguish between clouds C1, C2 and C3. Instead we will put a line now to separate them.

Finally, write $\text{SO}_2 < \text{DL}$ not “SO2 too low”

Done

What are the experimental conditions in figure 4: pressure temperature etc....?

In the main text we write that this is based on: “NCAR TUV calculator for 25 km altitude at the coordinates of the HCHO enhancement for Jan-15 to Jan-17, reduced to 58% of the rate to represent conditions inside the plume, as modelled by Zhu et al. (25;Zhu2023)”.

Based on your comment, we will also clarify this in the figure caption: “HCHO in-plume photolysis rate as a function of time (solid line), starting from the moment of the eruption (based on NCAR TUV calculator).”

Could you find any TROPOMI overpasses over land where you can also detect CH₄ in the plume? E.g., over Australia? This would be a nice addition to the paper.

We added: “This corresponds with a CH₄ vertical column density enhancement of 0.18×10^{19} molec/cm², which is around 4% of a typical background measured by TROPOMI of 4.2×10^{19} molec/cm². By Jan-20 the CH₄ concentration enhancement is expected to have dispersed by an order of magnitude, in line with the observed lower HCHO concentration enhancement. The methane enhancement is therefore also too low to detect with TROPOMI.”

L290 – Also noted in Evan et al., 2023.

We refer to Evan in this sentence.

L294 – Explain why this is surprising

We write: “This is surprising, because other researchers found a slight depletion of HCl compared to seasonal average (27;Evan2023).”

L298 – What version is the HO₂ MLS data and does it pass QA? If not, it should likely not be used without guidance from the MLS team.

We used MLS v5, added to the text now: “with coincidental simultaneous MLS v5 observations”

The Jan-21 HO₂ data that we used for our conclusions and that is shown in the added figure has passed QA.

In addition, if we would use modelled HO₂ instead of MLS observations, it means a 6x lower Cl production via HOBr and HOCl, so would not change the conclusions of the paper (we will add a note about this in the relevant text in the SI).

L303 – “can only be” replace with “is most likely”

Our argumentation in this paragraph leaves no doubt that HCHO must be continuously produced to explain the observation. Therefore we believe “can only be explained by in-plume production” is accurate.

L355-363 Overly negative and at the same time vague – unpack the differences in chemical mechanisms.

Changes made to the paragraph: we added ‘a mechanism of HOCl uptake resulting in’ and ‘(we discuss this mechanism later in the discussion)’.

The mechanism in Zhu is the HOCl uptake mechanism that we unpack in a later paragraph. In the specific sentences you refer to we are comparing rates to show there is disagreement. We certainly don’t want to be negative, because our paper is built on the high quality work by Zhu. We don’t understand why you find this paragraph ‘overly negative’ (otherwise we would rephrase).

L359 – (rate is not observed but inferred based on relationship to HCHO and other variables).

Changed to ‘inferred’

L400 – Check the date – do you mean “Jan. 16” instead of “Feb. 16”?

You are correct – changed.

L443 – Why are there two discussion sections? (previously also L301)

We apologize for the confusion. The second 'discussion' section is meant to be a 'conclusion' section, and this has now been adapted.

L475 – but not HO₂ (from MLS why not?)

We don't know why HO₂ was not discussed in existing scientific literature on HHTH. Instead, other studies inferred HO₂ from modelling. As mentioned, if we would use modeled HO₂ concentrations by Zhu, they would be lower and our conclusions on Cl production by HOBr and HOCl would be even stronger.

L468 – 481 This entire section belongs in introduction. There is no mention of the Hunga eruption in the introduction which seems odd considering it is central to the paper.

We have moved this HHTH section to the introduction, and deleted one paragraph on methane removal interventions ~~“A recently proposed approach to atmospheric methane removal~~“

L481 – One of the other most interesting features of the Hunga eruption was the rapid conversion of SO₂ to aerosol (Legras 2022; Asher et al., 2023) which can be explained by elevated OH in the water rich plume...

Your comment raises an interesting question that we have added to the discussion:

~~“An interesting feature of the HHTH eruption was the rapid conversion of SO₂ to aerosol (Legras 2022; Asher et al., 2023) which can be explained by elevated OH in the water rich plume. Such an OH enhancement could result from enhanced methane oxidation by chlorine (Pennacchio2024). However, our observed high Cl production raises the question whether the rapid SO₂ conversion is due to chlorine oxidation instead? Studies of SO₂ oxidation by sea-spray aerosols provide evidence that chlorine can cause SO₂ oxidation, especially through HOCl (Harris2012).”~~

Harris, E., Sinha, B., Hoppe, P., Foley, S., and Borrmann, S.: Fractionation of sulfur isotopes during heterogeneous oxidation of SO₂ on sea salt aerosol: a new tool to investigate non-sea salt sulfate production in the marine boundary layer, Atmos. Chem. Phys., 12, 4619–4631, <https://doi.org/10.5194/acp-12-4619-2012>, 2012.

As mentioned above, with an abundance of OH, wouldn't CH₄ be rapidly oxidized even without Cl? If not, please explain.

See our response to your question on OH, and the added text in the manuscript, above. In short, the rapid SO₂ oxidation would need OH levels that are 6x enhanced (based on the lifetime calculations by Asher), and would be lower than the maximum of 1e7 /cm³ that we used in the added text. Such an OH concentration would require a too high CH₄ emission by the volcano to explain the observed HCHO observation.

We refer to previous added text: ~~“If 100% of this is due to a maximum OH enhancement of 1 × 10⁷ cm⁻³, the CH₄ concentration was at least 95 ppm compared to a background value of 1 ppm (an enhancement of at least 2300 Gg CH₄).” and “This rules out the possibility that the HCHO enhancement is due to OH (it would mean at least 2300 Gg CH₄ emission that would be clearly visible on ACE-FTS).”~~

L548 – Methodologies – you should show equations for clarity (even if they are very simple).

Added to method 1: $E_{tot} = \sum_{i=1}^N (HCHO_i - HCHO_{i,baseline}) A_i$, in which $HCHO_i$ is the HCHO VCD for pixel i , $HCHO_{i,baseline}$ is the baseline value, A_i is the surface area represented by pixel i , and N is the number of pixels in the area of interest.

And for method 2: $E_{tot} = \frac{\Delta HCHO}{\Delta X} \sum_{i=1}^N (X_i - X_{i,baseline}) A_i$, where $X_{i,baseline}$ is 0 for SO_2 and AOD.

Also, Method 2 you say is more accurate but it would seem to introduce new uncertainty to the estimate (based on the uncertainty in other data).

We write: “provides a more accurate quantification by combining additional data.”

L560- 568 How is the empirical AMF correction derived? We need more details on this or a citation.

We clarified this better now and added a reference: “Instead of an AMF of 1.3 for tropospheric background HCHO, we used the altitude-resolved air mass factors of the HCHO retrievals (DeSmedt2018), and took a value for a stratospheric layer around 25 km of 6.3 (correction factor 4.85).”

References

[1]

Y. Zhu et al., “Perturbations in stratospheric aerosol evolution due to the water-rich plume of the 2022 Hunga-Tonga eruption,” *Commun Earth Environ*, vol. 3, no. 1, p. 248, Oct. 2022, doi: sc.

[2]

E. Asher et al., “Unexpectedly rapid aerosol formation in the Hunga Tonga plume,” *Proc. Natl. Acad. Sci. U.S.A.*, vol. 120, no. 46, p. e2219547120, Nov. 2023, doi: 10.1073/pnas.2219547120.